# Deep Clustering and Representation Learning that Preserves Geometric Structures

## Abstract

In this paper, we propose a novel framework for Deep Clustering and multi-manifold Representation Learning (DCRL) that preserves the geometric structure of data. In the proposed DCRL framework, manifold clustering is done in the latent space guided by a clustering loss. To overcome the problem that clustering-oriented losses may deteriorate the geometric structure of embeddings in the latent space, an isometric loss is proposed for preserving intra-manifold structure locally and a ranking loss for inter-manifold structure globally. Experimental results on various datasets show that the DCRL framework leads to performances comparable to current state-of-the-art deep clustering algorithms, yet exhibits superior performance for manifold representation. Our results also demonstrate the importance and effectiveness of the proposed losses in preserving geometric structure in terms of visualization and performance metrics. The code is provided in the Supplementary Material.

## 1 Introduction

Clustering, a fundamental tool for data analysis and visualization, has been an essential research topic in data science and machine learning. Conventional clustering algorithms such as $K$-Means (MacQueen, 1965), Gaussian Mixture Models (GMM) (Bishop, 2006), and spectral clustering (Shi & Malik, 2000) perform clustering based on distance or similarity. However, handcrafted distance or similarity measures are rarely reliable for large-scale high-dimensional data, making it increasingly challenging to achieve effective clustering. An intuitive solution is to transform the data from the high-dimensional input space to the low-dimensional latent space and then to cluster the data in the latent space. This can be achieved by applying dimensionality reduction techniques such as PCA (Wold et al., 1987), t-SNE (Maaten & Hinton, 2008), and UMAP (McInnes et al., 2018). However, since these methods are not specifically designed for clustering tasks, some of their properties may be contrary to our expectations, e.g., two data points from different manifolds that are close in the input space will be closer in the latent space derived by UMAP. Therefore, the first question here is *how to learn the manifold representation that favors clustering?*

The two main points for the multi-manifold representation learning are *Point (1)* preserving the local geometric structure within each manifold and *Point (2)* ensuring the discriminability between different manifolds. Most previous work seems to start with the assumption that the label of each data point is known, and then design the algorithm in a *supervised manner*, which greatly simplifies the problem of multi-manifold learning. However, it is challenging to decouple complex cross-over relations and ensure discriminability between different manifolds, especially in *unsupervised* settings. One natural strategy is to achieve *Point (2)* through performing clustering in the input space to get pseudo-labels and then performing representation learning for each manifold. However, clustering is in fact contradictory to *Point (1)* (which will be analyzed in detail in Sec. 3.3), making it important to alleviate this contradiction so that clustering helps both *point (1)* and *point (2)*. Thus, the second question here is *how to cluster data that favors learning manifold representation?*

To answer these two questions, some pioneering work has proposed to integrate deep clustering and representation learning into a unified framework by defining a clustering-oriented loss. Though promising performance has been demonstrated on various datasets, we observe that a vital factor has been ignored by these work that the defined clustering-oriented loss may deteriorate the geometric

structure of the latent space [1], which in turn hurts the performance of visualization, clustering generalization, and manifold representation. In this paper, we propose to jointly perform deep clustering and multi-manifold representation learning with geometric structure preservation. Inspired by Xie et al. (2016), the clustering centers are defined as a set of **learnable** parameters, and we use a clustering loss to simultaneously guide the separation of data points from different manifolds and the learning of the clustering centers. To prevent clustering loss from deteriorating the latent space, an isometric loss and a ranking loss are proposed to preserve the intra-manifold structure locally and inter-manifold structure globally. Finally, we achieve the following three goals related to clustering, geometric structure, and manifold representation: (1) Clustering helps to ensure inter-manifold discriminability; (2) Local structure preservation can be achieved with the presence of clustering; (3) Geometric structure preservation helps clustering.

The contributions of this work are summarized as below:

- Proposing to integrate deep clustering and multi-manifold representation learning into a unified framework with local and global structure preservation.
- Unlike conventional multi-manifold learning algorithms that deal with all point pair relationships between different manifolds simultaneously, we set the clustering centers as a set of learnable parameters and achieve global structure preservation in a faster, more efficient, and easier to optimize manner by applying ranking loss to the clustering centers.
- Analyzing the contradiction between two optimization goals of clustering and local structure preservation and proposing an elegant training strategy to alleviate it.
- The proposed DCRL algorithm outperforms competing algorithms in terms of clustering effect, generalizability to out-of-sample, and performance in manifold representation.

## 2  RELATED WORK

**Clustering analysis**. As a fundamental tool in machine learning, it has been widely applied in various domains. One branch of classical clustering is $K$-Means (MacQueen, 1965) and Gaussian Mixture Models (GMM) (Bishop, 2006), which are fast, easy to understand, and can be applied to a large number of problems. However, limited by Euclidean measure, their performance on high-dimensional data is often unsatisfactory. Spectral clustering and its variants (such as SC-Ncut (Bishop, 2006)) extend clustering to high-dimensional data by allowing more flexible distance measures. However, limited by the computational efficiency of the full Laplace matrix, spectral clustering is challenging to extend to large-scale datasets.

**Deep clustering**. The success of deep learning has contributed to the growth of *deep clustering*. One branch of deep clustering performs clustering after learning a representation through existing unsupervised techniques. For example, Tian et al. (2014) use autoencoder to learn low dimensional features and then run $K$-Means to get clustering results (AE+$K$-Means). Considering the geometric structure of the data, N2D (McConville et al., 2019) applies UMAP to find the best clusterable manifold of the obtained embedding, and then run $K$-Means to discover higher-quality clusters. The other category of algorithms tries to optimize clustering and representation learning jointly. The closest work to us is Deep Embedding Clustering (DEC) (Xie et al., 2016), which learns a mapping from the input space to a low dimensional latent space through iteratively optimizing clustering-oriented objective. As a modified version of DEC, while IDEC (Guo et al., 2017) claims to preserve the local structure of the data, in reality, their contribution is nothing more than adding a reconstruction loss. JULE (Yang et al., 2016b) unifies unsupervised representation learning with clustering based on the CNN architecture to improve clustering accuracy, which can be considered as a neural extension of hierarchical clustering. DSC devises a dual autoencoder to embed data into latent space, and then deep spectral clustering (Shaham et al., 2018) is applied to obtain label assignments (Yang et al., 2019). ASPC-DA (Guo et al., 2019) combines data augmentation with self-paced learning to encourage the learned features to be cluster-oriented. While sometimes they both evaluate performance in terms of accuracy, we would like to highlight that *deep clustering and visual self-supervised learning (SSL)* are two different research fields. SSL typically uses more powerful CNN architecture (applicable only to image data), and uses sophisticated techniques such as contrastive learning (He

---

[1]This claim was first made by IDEC (Guo et al., 2017), but they did not provide experiments to support it. In this paper, however, we show that the geometry of the latent space is indeed disrupted by visualization of learned embeddings (Fig. 4), visualization of clustering process (Fig. A3), and statistical analysis (Fig. A5).

et al., 2020), data augmentation (Chen et al., 2020), and clustering (Zhan et al., 2020; Ji et al., 2019; Van Gansbeke et al., 2020) for better performance on large-scale datasets such as ImageNet. Deep clustering, however, uses general MLP architecture (applicable to both image and vector data), so it is difficult to scale directly to large datasets without considering those sophisticated techniques.

**Manifold Representation Learning**. Isomap, as a representative algorithm of *single-manifold* learning, aims to capture global nonlinear features and seek an optimal subspace that best preserves the geodesic distance between data points (Tenenbaum et al., 2000). In contrast, some algorithms, such as the Locally Linear Embedding (LLE) (Roweis & Saul, 2000), are more concerned with the preservation of local neighborhood information. Combining DNN with manifold learning, the recently proposed Markov-Lipschitz Deep Learning (MLDL) algorithm achieves the preservation of local and global geometries by imposing Locally Isometric Smoothness (LIS) prior constraints (Li et al., 2020). Furthermore, *multi-manifold* learning is proposed to obtain intrinsic properties of different manifolds. Yang et al. (2016a) proposed a supervised discriminant isomap where data points are partitioned into different manifolds according to label information. Similarly, Zhang et al. (2018) proposed a semi-supervised learning framework that applies the labeled and unlabeled training samples to perform the joint learning of local neighborhood-preserving features. In most previous work on multi-manifold learning, the problem is considered from the perspective that the label is known or partially known, which significantly simplifies the problem. However, it is challenging to decouple multiple overlapping manifolds in unsupervised settings, and that is what this paper aims to explore.

## 3 PROPOSED METHOD

Consider a dataset $X$ with $N$ samples, and each sample $x_i \in \mathbb{R}^d$ is sampled from $C$ different manifolds $\{M_c\}_{c=1}^C$. Assume that each category in the dataset lies in a compact low-dimensional manifold, and the number of manifolds $C$ is prior knowledge. Define two nonlinear mapping $z_i = f(x_i, \theta_f)$ and $y_i = g(z_i, \theta_g)$, where $z_i \in \mathbb{R}^m$ is the embedding of $x_i$ in the latent space, $y_i$ is the reconstruction of $x_i$. The $j$-th cluster center is denoted as $\mu_j \in \mathbb{R}^m$, where $\{\mu_j\}_{j=1}^C$ is defined as a set of **learnable** parameters. We aim to find optimal parameters $\theta_f$ and $\mu$ so that the embeddings $\{z_i\}_{i=1}^N$ can achieve clustering with local and global structure preservation. To this end, a denoising autoencoder (Vincent et al., 2010) shown in Fig 1 is first pre-trained in an unsupervised manner to learn an initial latent space. Denoising autoencoder aims to optimize the self-reconstruction loss $L_{AE} = MSE(\hat{x}, y)$, where the $\hat{x}$ is a copy of $x$ with Gaussian noise added, that is, $\hat{x} = x + N(0, \sigma^2)$. Then the autoencoder is finetuned by optimizing the following clustering-oriented loss $\{L_{cluster}(z, \mu)\}$ and structure-oriented losses $\{L_{rank}(x, \mu), L_{LIS}(x, z), L_{align}(z, \mu)\}$. Since the clustering should be performed on features of **clean** data, instead of noised data $\hat{x}$ that is used in denoising autoencoder, the clean data $x$ is used for fine-tuning.

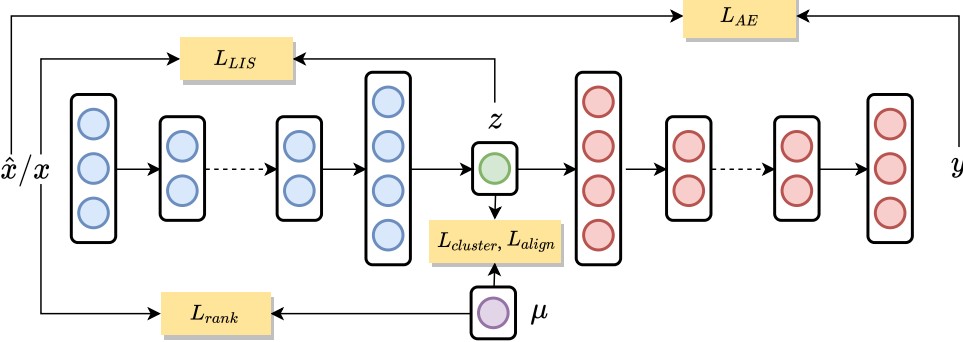

Figure 1: The framework of the proposed DCRL method. The encoder, decoder, latent space, and cluster centers are marked as blue, red, green, and purple, respectively.

### 3.1 Clustering-oriented Loss

First, the cluster centers $\{\mu_j\}_{j=1}^{C}$ in the latent space $Z$ are initialized (the initialization method will be introduced in Sec 4.1). Then the similarity between the embedded point $z_i$ and cluster centers $\{\mu_j\}_{j=1}^{C}$ is measured by Student's $t$-distribution:

$$q_{ij} = \frac{\left(1 + \|z_i - \mu_j\|^2\right)^{-1}}{\sum_{j'} \left(1 + \|z_i - \mu_{j'}\|^2\right)^{-1}} \tag{1}$$

The auxiliary target distribution is designed to help manipulate the latent space, defined as:

$$p_{ij} = \frac{q_{ij}^2/f_j}{\sum_{j'} q_{ij'}^2/f_{j'}}, \quad where \quad f_j = \sum_i q_{ij} \tag{2}$$

where $f_j$ is the normalized cluster frequency, used to balance the size of different clusters. Then the encoder is optimized by the following objective:

$$L_{cluster} = \text{KL}(P\|Q) = \sum_i \sum_j p_{ij} \log \frac{p_{ij}}{q_{ij}} \tag{3}$$

The gradient of $L_{cluster}$ with respect to each learnable cluster center $\mu_j$ can be computed as:

$$\frac{\partial L_{cluster}}{\partial \mu_j} = -\sum_i \left(1 + \|z_i - \mu_j\|^2\right)^{-1} \cdot (p_{ij} - q_{ij})(z_i - \mu_j) \tag{4}$$

$L_{cluster}$ facilitates the aggregation of data points within the same manifold, while data points from different manifolds are kept away from each other. However, we find that the clustering-oriented loss may deteriorate the geometric structure of the latent space, which hurts the clustering accuracy and leads to meaningless representation. To prevent the deterioration of clustering loss, we introduce isometry loss $L_{LIS}$ and ranking loss $L_{rank}$ to preserve the local and global structure, respectively.

### 3.2 Structure-oriented Loss

**Intra-manifold Isometry Loss.** The intra-manifold local structure is preserved by optimizing the following objective:

$$L_{LIS} = \sum_{i=1}^{N} \sum_{j \in \mathcal{N}_i^Z} |d_X(x_i, x_j) - d_Z(z_i, z_j)| \cdot \pi(l(x_i) = l(x_j)) \tag{5}$$

where $\mathcal{N}_i^Z$ represents the neighborhood of data point $z_i$ in the feature space $Z$, and the $k$NN is applied to determine the neighborhood. $\pi(\cdot) \in \{0, 1\}$ is an indicator function, and $l(x_i)$ is a manifold determination function that returns the manifold $s_i$ where sample $x_i$ is located, that is $s_i = l(x_i) = \arg\max_j p_{ij}$. Then we can derive $C$ manifolds $\{M_c\}_{c=1}^{C}$: $M_c = \{x_i; \ s_i = c, i = 1, 2, ..., N\}$. In a nutshell, the loss $L_{LIS}$ constrains the isometry within each manifold.

**Inter-manifold Ranking Loss.** The inter-manifold global structure is preserved by optimizing the following objective:

$$L_{rank} = \sum_{i=1}^{C} \sum_{j=1}^{C} \left| d_Z(\mu_i, \mu_j) - \kappa \cdot d_X\left(v_i^X, v_j^X\right) \right| \tag{6}$$

where $\{v_j^X\}_{j=1}^{C}$ is defined as the centers of different manifolds in the original input space $X$ with $v_j^X = \frac{1}{|M_j|} \sum_{i \in M_j} x_i$ $(j = 1, 2, ..., C)$. The parameter $\kappa$ determines the extent to which different manifolds move away from each other. The larger $\kappa$ is, the further away the different manifolds are from each other. The derivation for the gradient of $L_{rank}$ with respect to each learnable cluster center $\mu_j$ is placed in **Appendix A.1**. Note that $L_{rank}$ is optimized in an iterative manner, rather than by initializing $\{\mu_j\}_{j=1}^{C}$ once and then separating different clusters based only on the initialization

results. Additionally, contrary to us, the conventional methods for dealing with inter-manifold separation typically impose push-away constraints on all data points from different manifolds (Zhang et al., 2018; Yang et al., 2016a), defined as:

$$L_{sep} = -\sum_{i=1}^{N} \sum_{j=1}^{N} d_Z(z_i, z_j) \cdot \pi(l(x_i) \neq l(x_j)) \tag{7}$$

The main differences between $L_{rank}$ and $L_{sep}$ are as follows: (1) $L_{sep}$ imposes constraints on embedding points $\{z_i\}_{i=1}^{N}$, which in turn indirectly affects the network parameters $\theta_f$. In contrast, $L_{rank}$ imposes rank-preservation constrains directly on learnable parameters $\{\mu_j\}_{j=1}^{C}$ in the form of *regularization* item to control the separation of the clustering centers. (2) $L_{rank}$ is easier to optimize, faster to process, and more accurate. $L_{sep}$ is imposed on all data points from different manifolds, which involves $N \times N$ *point-to-point* relationships. This means that each point may be subject to the push-away force from other manifolds, but at the same time, each point has to meet the isometry constraint with its neighboring points. Under these two constraints, optimization is difficult and it is easy to fall into a local optimal solution and output inaccurate results. In contrast, $L_{rank}$ is imposed directly on the clustering centers, involving only $C \times C$ *cluster-to-cluster* relationships, which avoids the above problem and makes it easier to optimize. (3) The parameter $\kappa$ introduced in $L_{rank}$ allows us to control the extent of separation between manifolds for specific downstream tasks.

**Alignment Loss.** Note that the global ranking loss $L_{rank}$ is imposed directly on the learnable parameter $\{\mu_j\}_{j=1}^{C}$, so optimizing $L_{rank}$ only updates $\{\mu_j\}_{j=1}^{C}$ rather the encoder's parameter $\theta_f$. However, the optimization of $\{\mu_j\}_{j=1}^{C}$ not only relies on $L_{rank}$, but is also constrained by $L_{cluster}$, which ensures that data points remain roughly distributed around cluster centers and do not deviate significantly from them during the optimization process. Alignment loss $L_{align}$, as an auxiliary term, aims to help align learnable cluster centers $\{\mu_j\}_{j=1}^{C}$ with real cluster centers $\{v_j^Z\}_{j=1}^{C}$ and make this binding stronger:

$$L_{align} = \sum_{j=1}^{C} ||\mu_j - v_j^Z|| \tag{8}$$

where $\{v_j^Z\}_{j=1}^{C}$ are defined as $v_j^Z = \frac{1}{|M_j|} \sum_{i \in M_j} z_i$ ($j = 1, 2, ..., C$). The derivation for the gradient of $L_{align}$ with respect to each learnable cluster center $\mu_j$ is placed in **Appendix A.1**.

### 3.3 TRAINING STRATEGY

#### 3.3.1 CONTRADICTION

The contradiction between clustering and local structure preservation is analyzed from the *forces analysis* perspective. As shown in Fig 2, we assume that there exists a data point (red point) and its three nearest neighbors (blue points) around a cluster center (gray point). When clustering and local structure preserving are optimized simultaneously, it is very easy to fall into a local optimum, where the data point is in steady-state, and the resultant force from its three nearest neighbors is equal in magnitude and opposite to the gravitational forces of the cluster. Therefore, the following training strategy is applied to prevent such local optimal solutions.

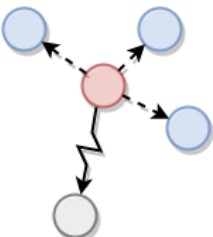

Figure 2: Force analysis of the contradiction between clustering and local structure preservation.

### 3.3.2 ALTERNATING TRAINING AND WEIGHT CONTINUATION

**Alternating Training.** To solve the above problem and integrate the goals of clustering and structure preservation into a unified framework, we take an alternating training strategy. Within each epoch, we first jointly optimize $L_{cluster}$ and $L_{rank}$ in a *mini-batch*, with joint loss defined as

$$L_1 = L_{AE} + \alpha L_{cluster} + L_{rank} \qquad (9)$$

where $\alpha$ is the weighting factor that balances the effects of clustering and global rank-preservation. Then at each epoch, we optimize isometry loss $L_{LIS}$ and $L_{align}$ on *the whole dataset*, defined as

$$L_2 = \beta L_{LIS} + L_{align} \qquad (10)$$

**Weight continuation.** At different stages of training, we have different expectations for the clustering and structure-preserving. At the beginning of training, to successfully decouple the overlapping manifolds, we hope that the $L_{cluster}$ will dominate and $L_{LIS}$ will be auxiliary. When the margin between different manifolds is sufficiently pronounced, the weight $\alpha$ for $L_{cluster}$ can be gradually reduced, while the weight $\beta$ for $L_{LIS}$ can be gradually increased, focusing on the preservation of the local isometry. The whole algorithm is summarized in Algorithm 1 in **Appendix A.2**.

**Three-stage explanation.** The training process can be roughly divided into three stages, as shown in Fig 3, to explain the training strategy more vividly. At first, four different manifolds overlap. At Stage 1, $L_{cluster}$ dominates, thus data points within each manifold converge towards cluster centers to form spheres, but the local structure of manifolds is destroyed. At Stage 2, $L_{rank}$ dominates, thus different manifolds in the latent space move away from each other to increase the manifold margin and enhance the discriminability. At stage 3, the manifolds gradually recover their original local structure from the spherical shape with $L_{LIS}$ dominating. It is worth noting that the above losses may coexist with each other rather than being completely independent at different stages, but that the role played by different losses varies due to the alternating training and weight continuation.

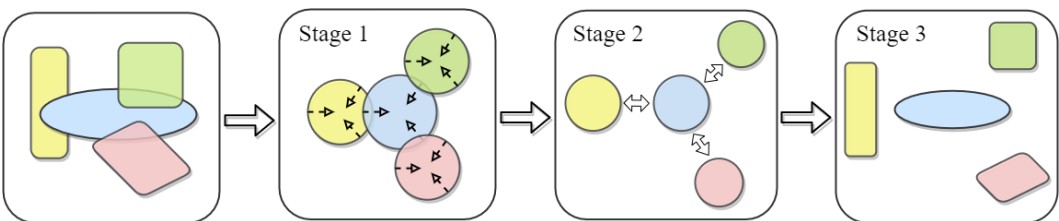

Figure 3: Schematic of training strategy. Four different colors and shapes represent four intersecting manifolds, and three stages involve the clustering, separation, and structure recovery of manifolds.

## 4 EXPERIMENTS

### 4.1 EXPERIMENTAL SETUPS

In this section, the effectiveness of the proposed framework is evaluated in 6 benchmark datasets: MNIST-full, MNIST-test, USPS, Fashion-MNIST, REUTERS-10K and HAR, on which our method is compared with 9 other methods mentioned in Sec 2 in 8 evaluation metrics including metrics designed specifically for clustering and manifold representation learning. The brief descriptions of the datasets are given in **Appendix A.3**.

**Parameters settings.** Currently we use MLP architecture for this version and will extend it to ConvAE in the future. The encoder structure is $d$-500-500-500-2000-10 where $d$ is the dimension of the input data, and the decoder is its mirror. After pretraining, in order to initialize the learnable clustering centers, the t-SNE is applied to transform the latent space $Z$ to 2 dimensions further, and then the $K$-Means algorithm is run to obtain the label assignments for each data point [2]. The centers of each category in the latent space $Z$ are set as initial cluster centers $\{\mu_j\}_{j=1}^C$. The batch size is set to 256, the epoch is set to 300, the parameter $k$ for nearest neighbor is set to 5, and the parameter $\kappa$

---

[2]Since cluster centers $\{\mu_j\}_{j=1}^C$ are **learnable** and updated in an iterative manner, we believe that a proper initialization is sufficient, and the exploration of initialization methods is beyond the scope of this paper.

is set to 3 for all datasets. Sensitivity analysis for parameters $k$ and $\kappa$ is available in **Appendix A.12**. Besides, Adam optimizer (Kingma & Ba, 2014) with learning rate $\lambda$=0.001 is used. As described in Sec 3.3.2, the weight continuation is applied to train the model. The weight parameter $\alpha$ for $L_{cluster}$ decreases linearly from 0.1 to 0 within epoch 0-150. In contrast, the weight parameter $\beta$ for $L_{LIS}$ increases linearly from 0 to 1.0 within epoch 0-150. In this paper, each set of experiments is run 5 times with different 5 random seeds, and the results are averaged into the final performance metrics. The implementation uses the PyTorch library running on NVIDIA v100 GPU.

**Evaluation Metrics.** Two standard evaluation metrics: Accuracy (ACC) and Normalized Mutual Information (NMI) (Xu et al., 2003) are used to evaluate clustering performance. Besides, six evaluation metrics are adopted in this paper to evaluate the performance of multi-manifold representation learning, including Relative Rank Error (RRE), Trustworthiness (Trust), Continuity (Cont), Root Mean Reconstruction Error (RMRE), Locally Geometric Distortion (LGD) and Cluster Rank Accuracy (CRA). Limited by space, their precise definitions are available in **Appendix A.4**.

## 4.2 EVALUATION OF CLUSTERING

### 4.2.1 QUANTITATIVE COMPARISON

The metrics ACC/NMI of different methods on various datasets are reported in Tab 1. For those comparison methods whose results are not reported or the experimental settings are not clear on some datasets, we run the released code using the hyperparameters provided in their paper with *the same random seeds and initialization*, then report their *average* performance, and label them with (*). While ASPC-DA achieves the best performance on three datasets (MNIST-test, MNIST-full, and USPS), its performance gains do not come directly from clustering, but from sophisticated modules such as data augmentation and self-paced learning. Once these modules are removed, there is a very large degradation in its performance. For example, with data augmentation removed, ASPC-DA achieves less competitive performance, e.g., an accuracy of 0.931 (*vs* 0.988) on MNIST-full, 0.813 (*vs* 0.973) on MNIST-test and 0.768 (*vs* 0.982) on USPS. Though ASPC-DA is based on the MLP architecture, its image-based Data Augmentation (DA) cannot be applied directly to vector data, which explains why ASPC has no performance advantage on the vector-based REUTERS-10K and HAR datasets (even compared to DEC and IDEC).

In a fairer comparison (without considering ASPC-DA), we find that DCRL outperforms $K$-Means and SC-Ncut by a significant margin and surpasses the other seven competing DNN-based algorithms on all datasets except MNIST-test. Even with the MNIST-test dataset, we still rank second, outperforming the third by 1.1%. In particular, we obtained the best performance on the Fashion-MNIST and HAR (vector) dataset , and more notably, our clustering accuracy exceeds the current SOTA method by 5.1% and 4.9%, respectively.

Table 1: Clustering performance (ACC/NMI) of different algorithms on six datasets.

| Algorithms | MNIST-full | MNIST-test | USPS | Fashion-MNIST | REUTERS-10K | HAR |
|---|---|---|---|---|---|---|
| $K$-Means | 0.532/0.500 | 0.546/0.501 | 0.668/0.601 | 0.474/0.512 | 0.599/0.375* | 0.599/0.588 |
| SC-Ncut | 0.656/0.731 | 0.660/0.704 | 0.649/0.794 | 0.508/0.575 | 0.658/0.401* | 0.538/0.741 |
| AE+$K$-Means | 0.818/0.747 | 0.815/0.784* | 0.662/0.693 | 0.566/0.585* | 0.721/0.432* | 0.674/0.670* |
| DEC* | 0.903/0.854* | 0.885/0.851* | 0.889/0.873* | 0.554/0.576* | 0.773/0.528* | 0.759/0.695* |
| IDEC* | 0.918/0.868* | 0.876/0.817* | 0.893/0.876* | 0.572/0.601* | 0.785/0.541* | 0.786/0.718* |
| JULE | 0.964/0.913 | 0.961/0.915 | 0.950/0.913 | 0.563/0.608 | - | - |
| DSC | 0.978/0.941 | **0.980/0.946** | 0.869/0.857 | 0.662/0.645 | - | - |
| ASPC-DA | 0.988/0.966 | 0.973/0.936 | 0.982/0.951 | 0.591/0.654 | - | - |
| ASPC (w/o DA) | 0.931/0.886* | 0.813/0.792* | 0.768/0.803* | 0.576/0.632* | 0.692/0.418* | 0.769/0.682* |
| N2D* | 0.969/0.928* | 0.954/0.897* | 0.954/0.898* | 0.671/0.678* | 0.784/0.536* | 0.796/0.721* |
| DCRL (ours) | **0.980/0.946** | 0.972/0.930 | **0.960/0.902** | **0.710/0.685** | **0.836/0.590** | **0.845/0.758** |

### 4.2.2 GENERALIZABILITY EVALUATION

Tab 2 demonstrates that a learned DCRL can generalize well to unseen data with high clustering accuracy. Taking MNIST-full as an example, DCRL was trained using 50,000 training samples and then tested on the remaining 20,000 testing samples using the learned model. In terms of the metrics ACC and MNI, our method is optimal for both training and testing samples. More importantly, there is hardly any degradation in the performance of our method on the testing samples compared

to the training samples, while all other methods showed a significant drop in performance, e.g., DEC from 84.1% to 74.8%. This demonstrates the importance of geometric structure preservation for good generalizability. The testing visualization available in **Appendix A.5** shows that DCRL still maintains clear inter-cluster boundaries even on the test samples, which demonstrates the great generalizability of our method.

Table 2: Generalizability evaluated by ACC/NMI.

| Algorithms | training samples | testing samples |
|---|---|---|
| AE+$K$-Means | 0.815/0.736 | 0.751/0.711 |
| DEC | 0.841/0.773 | 0.748/0.704 |
| IDEC | 0.845/0.860 | 0.826/0.842 |
| JULE | 0.958/0.907 | 0.921/0.895 |
| DSC | 0.975/0.939 | 0.969/0.921 |
| N2D | 0.974/0.930 | 0.965/0.911 |
| DCRL (ours) | **0.978/0.941** | **0.978/0.941** |

### 4.2.3 CLUSTERING VISUALIZATION

The visualization of DCRL with several comparison methods is shown in Fig 4 (visualized using UMAP). From the perspective of clustering, our method is much better than the other methods. Among all methods, only DEC, IDEC, and DCRL can hold clear boundaries between different clusters, while the cluster boundaries of the other methods are indistinguishable. Although DEC and IDEC can successfully separate different clusters, they group many data points from different classes into the same cluster. Most importantly, due to the use of the clustering-oriented loss, the embedding learned by algorithms such as DEC, IDEC, JULE, and DSC (especially DSC) tend to form spheres and disrupt the original topological structure. Instead, our method overcomes these problems and achieves almost perfect separation between different clusters while preserving the local and global structure.

Additionally, the embedding of latent space during the training process is visualized in **Appendix A.6**, which is highly consistent with the three-stage explanation mentioned in Sec 3.3.2, showing that clustering-oriented does indeed do deteriorate the local geometric structure of the latent space, and designed $L_{LIS}$ helps to recover it. In addition, in the above experiments, the cluster number $C$ is assumed to be a known prior (which is consistent with the assumptions of almost all deep clustering algorithms). Therefore, we provide an additional experiment to explore what happens when $C$ is larger than the number of true clusters. It is found that there exists splitting of the clusters, but the different categories still maintain clear boundaries and are not mixed together, somewhat similar to hierarchical clustering. See **Appendix A.7** for detailed experimental settings and analysis.

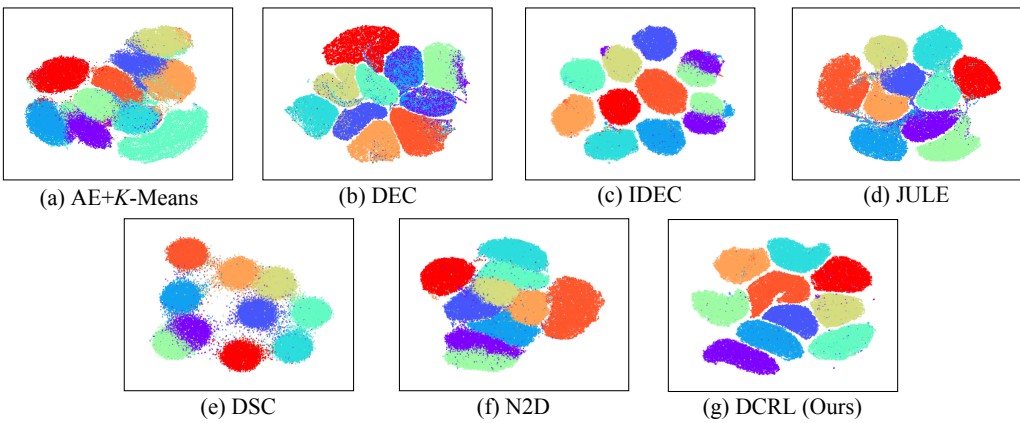

(a) AE+$K$-Means     (b) DEC     (c) IDEC     (d) JULE

(e) DSC     (f) N2D     (g) DCRL (Ours)

Figure 4: Visualization of the embeddings learned by different algorithms on MNIST-full dataset.

### 4.3 EVALUATION OF MULTI-MANIFOLD REPRESENTATION LEARNING

Although numerous previous work has claimed that they brought clustering and representation learning into a unified framework, they all, unfortunately, lack an analysis of the effectiveness of the learned representations. In this paper, we compare DCRL with the other five methods in six evaluation metrics on six datasets. (Limited by space, only MNIST-full results are provided in the Tab 3 and the complete results are in **Appendix A.8**). The results show that DCRL outperforms all other methods, especially in the CRA metric, which is not only the best on all datasets but also reaches 1.0. This means that the "rank" between different manifolds in the latent space is completely preserved and undamaged, which proves the effectiveness of our global ranking loss $L_{rank}$.

Moreover, statistical analysis is performed in this paper to show the extent to which local and global structure is preserved in the latent space for each algorithm. Limited by space, they are placed in **Appendix A.9**. Furthermore, we also evaluated whether the learned representations are meaningful through downstream tasks, and this experiment is available in **Appendix A.10**.

Table 3: Performance for multi-manifold representation learning.

| Methods | RRE | Trust | Cont | $d$-RMSE | LGD | CRA |
|---|---|---|---|---|---|---|
| DEC | 0.099 | 0.844 | 0.948 | 44.85 | 4.379 | 0.28 |
| IDEC | 0.009 | 0.998 | 0.979 | 24.58 | 1.714 | 0.33 |
| JULE | 0.026 | 0.936 | 0.983 | 28.34 | 2.129 | 0.27 |
| DSC | 0.097 | 0.873 | 0.925 | 6.98 | 1.198 | 0.23 |
| N2D | 0.010 | 0.992 | 0.984 | 5.71 | 0.699 | 0.21 |
| DCRL | **0.005** | **0.999** | **0.987** | **5.49** | **0.691** | **1.00** |

### 4.4 ABLATION STUDY

This evaluates the effects of the loss terms and training strategies in the DCRL with five sets of experiments: the model without (A) Structure-oriented Loss (SL); (B) Clustering-oriented Loss (CL); (C) Weight Continuation (WC); (D) Alternating Training (AT), and (E) the full model. Limited by space, only MNIST-full results are provided in Tab 4, and results for the other four datasets are in **Appendix A.11**. After analyzing the results, we can conclude: (1) CL is the most important factor for obtaining good clustering, the lack of which leads to unsuccessful clustering, hence the numbers in the table are not very meaningful and are shown in gray color. (2) SL not only brings subtle improvements in clustering but also greatly improves the performance of multi-manifold representation learning. (3) Our elegant training strategies (WC and AT) both improve the performance of clustering and multi-manifold representation learning to some extent, especially on metrics such as RRE, Trust, Cont, and CRA.

Table 4: Ablation study of loss items and training strategies on MNIST-full dataset.

| Datasets | Methods | ACC/NMI | RRE | Trust | Cont | $d$-RMSE | LGD | CRA |
|---|---|---|---|---|---|---|---|---|
| | w/o SL | 0.976/0.939 | 0.0093 | 0.9967 | 0.9816 | 24.589 | 1.6747 | 0.32 |
| | w/o CL | 0.814/0.736 | 0.0004 | 0.9998 | 0.9990 | 7.458 | 0.0487 | 1.00 |
| MNIST-full | w/o WC | 0.977/0.943 | 0.0065 | 0.9987 | 0.9860 | 5.576 | 0.6968 | 0.98 |
| | w/o AT | 0.978/0.944 | 0.0069 | 0.9986 | 0.9851 | 5.617 | 0.7037 | 0.96 |
| | full model | **0.980/0.946** | **0.0056** | **0.9997** | **0.9871** | **5.498** | **0.6916** | **1.00** |

## 5 CONCLUSION

The proposed DCRL framework imposes clustering-oriented and structure-oriented constraints to optimize the latent space for simultaneously performing clustering and multi-manifold representation learning with local and global structure preservation. Extensive experiments on image and vector datasets demonstrate that DCRL is not only comparable to the state-of-the-art deep clustering algorithms but also able to learn effective and robust manifold representation, which is beyond the capability of those clustering methods that only care about *clustering accuracy*. Future work will focus on the adaptive determination of manifolds (clusters) numbers and extend our work to CNN architecture for large-scale datasets.

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

APPENDIX

A.1 GRADIENT DERIVATION

In the paper, we have emphasized time and again that $\{\mu_j\}_{j=1}^C$ is a set of **learnalbe** parameters, which means that we can optimize it while optimizing the network parameter $\theta_f$. In Eq. (4) of the paper, we have presented the gradient of $L_{cluster}$ with respect to $\mu_j$. In addition to $L_{cluster}$, both $L_{rank}$ and $L_{align}$ are involving $\mu_j$. Hence, the detailed derivations for the gradient of $L_{rank}$ and $L_{align}$ with respect to $\mu_j$ are also provided. The gradient of $L_{rank}$ with respect to each learnalbe cluster center $\mu_j$ can be computed as:

$$\frac{\partial L_{rank}}{\partial \mu_j} = \frac{\partial \sum_{i'=1}^C \sum_{j'=1}^C \left| d_Z\left(\mu_{i'}, \mu_{j'}\right) - \kappa * d_X\left(v_{i'}^X, v_{j'}^X\right) \right|}{\partial \mu_j}$$

$$= \sum_{i'=1}^C \sum_{j'=1}^C \frac{\partial \left| d_Z\left(\mu_{i'}, \mu_{j'}\right) - \kappa * d_X\left(v_{i'}^X, v_j^X\right) \right|}{\partial \mu_j}$$

The Euclidean metric is used for both the input space and the hidden layer space, i.e., $d_Z\left(\mu_{i'}, \mu_{j'}\right) = \|\mu_{i'} - \mu_{j'}\|$. In addition, the symbols are somewhat abused for clear derivation, representing $\kappa * d_X\left(v_{i'}^X, v_{j'}^X\right)$ with $K$. Accordingly, Eq. (11) can be further derived as follows:

$$\frac{\partial L_{rank}}{\partial \mu_j} = \sum_{i'=1}^C \sum_{j'=1}^C \frac{\partial \left| d_Z\left(\mu_{i'}, \mu_{j'}\right) - \kappa * d_X\left(v_{i'}^X, v_{j'}^X\right) \right|}{\partial \mu_j}$$

$$= \sum_{i'=1}^C \sum_{j'=1}^C \frac{\partial \left| \|\mu_{i'} - \mu_{j'}\| - K \right|}{\partial \mu_j}$$

$$= \sum_{i'=1}^C \frac{\partial \left| \|\mu_{i'} - \mu_j\| - K \right|}{\partial \mu_j} + \sum_{j'=1}^C \frac{\partial \left| \|\mu_j - \mu_{j'}\| - K \right|}{\partial \mu_j}$$

$$= \sum_{i'=1}^C \frac{\partial \left( \|\mu_{i'} - \mu_j\| - K \right)}{\partial \mu_j} \cdot \frac{\|\mu_{i'} - \mu_j\| - K}{\left| \|\mu_{i'} - \mu_j\| - K \right|}$$

$$+ \sum_{j'=1}^C \frac{\partial \left( \|\mu_j - \mu_{j'}\| - K \right)}{\partial \mu_j} \cdot \frac{\|\mu_j - \mu_{j'}\| - K}{\left| \|\mu_j - \mu_{j'}\| - K \right|}$$

$$= \sum_{i'=1}^C \frac{\partial \|\mu_{i'} - \mu_j\|}{\partial \mu_j} \cdot \frac{\|\mu_{i'} - \mu_j\| - K}{\left| \|\mu_{i'} - \mu_j\| - K \right|}$$

$$+ \sum_{j'=1}^C \frac{\partial \|\mu_j - \mu_{j'}\|}{\partial \mu_j} \cdot \frac{\|\mu_j - \mu_{j'}\| - K}{\left| \|\mu_j - \mu_{j'}\| - K \right|}$$

$$= \sum_{i'=1}^C \frac{\mu_j - \mu_{i'}}{\|\mu_j - \mu_{i'}\|} \cdot \frac{\|\mu_j - \mu_{i'}\| - K}{\left| \|\mu_j - \mu_{i'}\| - K \right|} + \sum_{j'=1}^C \frac{\mu_j - \mu_{j'}}{\|\mu_j - \mu_{j'}\|} \cdot \frac{\|\mu_j - \mu_{j'}\| - K}{\left| \|\mu_j - \mu_{j'}\| - K \right|}$$

$$= 2 \sum_{i'=1}^C \frac{\mu_j - \mu_{i'}}{\|\mu_j - \mu_{i'}\|} \cdot \frac{\|\mu_j - \mu_{i'}\| - K}{\left| \|\mu_j - \mu_{i'}\| - K \right|}$$

$$= 2 \sum_{i'=1}^C \frac{\mu_j - \mu_{i'}}{\|\mu_j - \mu_{i'}\|} \cdot \frac{\|\mu_j - \mu_{i'}\| - \kappa * d_X\left(v_{i'}^X, v_j^X\right)}{\left| \|\mu_j - \mu_{i'}\| - \kappa * d_X\left(v_{i'}^X, v_j^X\right) \right|}$$

$$= 2 \sum_{i'=1}^C \frac{\mu_j - \mu_{i'}}{d_Z\left(\mu_j, \mu_{i'}\right)} \cdot \frac{d_Z\left(\mu_j, \mu_{i'}\right) - \kappa * d_X\left(v_{i'}^X, v_j^X\right)}{\left| d_Z\left(\mu_j, \mu_{i'}\right) - \kappa * d_X\left(v_{i'}^X, v_j^X\right) \right|}$$

The gradient of $L_{align}$ with respect to each learnalbe cluster center $\mu_j$ can be computed as:

$$
\begin{aligned}
\frac{\partial L_{align}}{\partial \mu_j} &= \frac{\partial \sum_{j'=1}^{C} ||\mu_{j'} - v_{j'}^Z||}{\partial \mu_j} \\
&= \sum_{j'=1}^{C} \frac{\partial ||\mu_{j'} - v_{j'}^Z||}{\partial \mu_j} \\
&= \frac{\partial ||\mu_j - v_j^Z||}{\partial \mu_j} \\
&= \frac{\partial (\mu_j - v_j^Z)}{\partial \mu_j} \cdot \frac{\mu_j - v_j^Z}{||\mu_j - v_j^Z||} \\
&= \frac{\mu_j - v_j^Z}{||\mu_j - v_j^Z||}
\end{aligned}
$$

## A.2 ALGORITHM

---
**Algorithm 1** Algorithm for Deep Clustering and Representation Learning
---
**Input:**
  Input samples: $X$; Number of clusters: $C$; Number of batches: $B$; Number of iterations: $E$.
**Output:**
  Autoencoder's weights: $\theta_f$ and $\theta_g$; Cluster labels $\{s_i\}_{i=1}^N$; Trainable cluster centers $\{\mu_j\}_{j=1}^C$.
1: Initialize the weight $\{\mu_j\}_{j=1}^C$, $\theta_f$ and $\theta_g$, and obtain initialized soft label assignment $\{s_i\}_{i=1}^N$.
2: **for** $epoch \in \{0,1,\cdots,E\}$ **do**
3:   Compute embedded points $\{z_i\}_{i=1}^N$ and distribution $Q$;
4:   Update target distribution $P$;
5:   Compute soft cluster centers $\{v_i^X\}_{i=1}^C$ and $\{v_i^Z\}_{i=1}^C$.
6:   **for** $batch \in \{0,1,\cdots,B\}$ **do**
7:     Pick up one batch of samples $X_{batch}$ from $X$;
8:     Compute corresponding distribution $Q_{batch}$ and it's reconstruction $Y_{batch}$;
9:     Pick up target distribution batch $P_{batch}$ from $P$;
10:     Compute loss $L_{ae}$, $L_{cluster}$ and $L_{rank}$;
11:     Update the weight $\theta_f$, $\theta_g$ and $\{\mu_j\}_{j=1}^C$.
12:   **end for**
13:   Compute $L_{iso}$ and $L_{align}$ on all samples;
14:   Update the weight $\theta_f$ and $\{\mu_j\}_{j=1}^C$;
15:   Assign new soft labels $\{s_i\}_{i=1}^N$.
16: **end for**
17: return $\theta_f$, $\theta_g$, $\{s_i\}_{i=1}^N$, $\{\mu_j\}_{j=1}^C$.

---

## A.3 DATASETS

To show that our method works well with various kinds of datasets, we choose the following six image and vector datasets. Some example images are shown in Fig A1, and the brief descriptions of the datasets are given in Tab A1.

- MNIST-full (LeCun et al., 1998): The MNIST-full dataset consists of 70,000 handwritten digits of $28 \times 28$ pixels. Each gray image is reshaped to a 784-dimensional vector.
- MNIST-test (LeCun et al., 1998): The MNIST-test is the testing part of the MNIST dataset, which contains a total of 10000 samples.
- USPS [3]: The USPS dataset is composed of 9298 gray-scale handwritten digit images with a size of 16x16 pixels.

---
[3]https://cs.nyu.edu/~roweis/data.html

Table A1: Description of Datasets.

| Dataset | Samples | Categories | Data Size |
|---|---|---|---|
| MNIST-full | 70000 | 10 | 28×28×1 |
| MNIST-test | 10000 | 10 | 28×28×1 |
| USPS | 9298 | 10 | 16×16×1 |
| Fashion-MNIST | 70000 | 10 | 28×28×1 |
| REUTERS-10K | 10000 | 4 | 2000 |
| HAR | 10299 | 6 | 561 |

- Fashion-MNIST (Xiao et al., 2017): This Fashion-MNIST dataset has the same number of images and the same image size as MNIST-full, but it is fairly more complicated. Instead of digits, it consists of various types of fashion products.

- REUTERS-10K: REUTERS (Lewis et al., 2004) is composed of around 810000 English news stories labeled with a category tree. Four root categories (corporate/industrial, government/social, markets, and economics) are used as labels and excluded all documents with multiple labels. Following DEC (Xie et al., 2016), a subset of 10000 examples are randomly sampled, and the tf-idf features on the 2000 most frequent words are computed. The sampled dataset is denoted REUTERS-10K.

- HAR: HAR is a time series dataset consisting of 10,299 sensor samples from a smartphone. It was collected from 30 people performing six different activities: walking, walking upstairs, walking downstairs, sitting, standing, and laying.

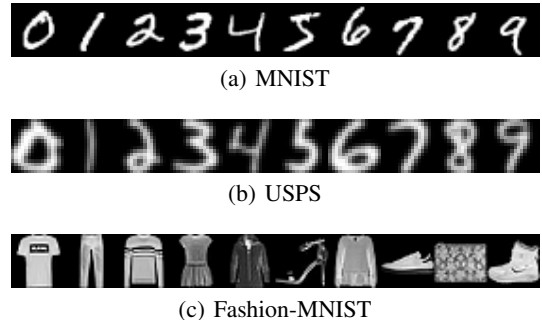

(a) MNIST

(b) USPS

(c) Fashion-MNIST

Figure A1: The image samples from three datasets (MNIST, USPS, and Fashion-MNIST)

### A.4 DEFINITIONS OF PERFORMANCE METRICS

The following notations are used for the definitions:

$d_X(i,j)$: the pairwise distance between $x_i$ and $x_j$ in input space $X$;

$d_Z(i,j)$: the pairwise distance between $z_i$ and $z_j$ in latent space $Z$;

$\mathcal{N}_i^{k,X}$: the set of indices to the $k$-nearest neighbor ($k$NN) of $x_i$ in input space $X$;

$\mathcal{N}_i^{k,Z}$: the set of indices to the $k$-nearest neighbor ($k$NN) of $z_i$ in latent space $Z$;

$r_X(i,j)$: the rank of the closeness (in Euclidean distance) of $x_j$ to $x_i$ in input space $X$;

$r_Z(i,j)$: the rank of the closeness (in Euclidean distance) of $z_j$ to $z_i$ in latent space $Z$.

The eight evaluation metrics are defined below:

(1) **ACC** (Accuracy) measures the accuracy of clustering:

$$ACC = \max_m \frac{\sum_{i=1}^N 1\{l_i = m(s_i)\}}{N}$$

where $l_i$ and $s_i$ are the true and predicted labels for data point $x_i$, respectively, and $m(\cdot)$ is all possible one-to-one mappings between clusters and label categories.

(2) **NMI** (Normalized Mutual Information) NMI calculates the normalized measure of similarity between two labels of the same data

$$NMI = \frac{I(l;s)}{\max\{H(l), H(s)\}}$$

where $I(l,s)$ is the mutual information between the real label $l$ and predicted label $s$, and $H(\cdot)$ represents their entropy.

(3) **RRE** (Relative Rank Change) measures the average of changes in neighbor ranking between two spaces $X$ and $Z$:

$$RRE = \frac{1}{(k_2 - k_1 + 1)} \sum_{k=k_1}^{k_2} \left\{ MR_{X \to Z}^k + MR_{Z \to X}^k \right\}$$

where $k_1$ and $k_2$ are the lower and upper bounds of the $k$-NN.

$$MR_{X \to Z}^k = \frac{1}{H_k} \sum_{i=1}^{N} \sum_{j \in \mathcal{N}_i^{k,Z}} \left( \frac{|r_X(i,j) - r_Z(i,j)|}{r_Z(i,j)} \right)$$

$$MR_{Z \to X}^k = \frac{1}{H_k} \sum_{i=1}^{N} \sum_{j \in \mathcal{N}_i^{k,X}} \left( \frac{|r_X(i,j) - r_Z(i,j)|}{r_X(i,j)} \right)$$

where $H_k$ is the normalizing term, defined as

$$H_k = N \sum_{l=1}^{k} \frac{|N - 2l|}{l}.$$

(4) **Trust** (Trustworthiness) measures to what extent the $k$ nearest neighbors of a point are preserved when going from the input space to the latent space:

$$Trust = \frac{1}{k_2 - k_1 + 1} \sum_{k=k_1}^{k_2} \left\{ 1 - \frac{2}{Nk(2N - 3k - 1)} \sum_{i=1}^{N} \sum_{j \in \mathcal{N}_i^{k,Z}, j \notin \mathcal{N}_i^{k,X}} (r_X(i,j) - k) \right\}$$

where $k_1$ and $k_2$ are the bounds of the number of nearest neighbors.

(5) **Cont** (Continuity) is defined analogously to $Trust$, but checks to what extent neighbors are preserved when going from the latent space to the input space:

$$Cont = \frac{1}{k_2 - k_1 + 1} \sum_{k=k_1}^{k_2} \left\{ 1 - \frac{2}{Nk(2N - 3k - 1)} \sum_{i=1}^{N} \sum_{j \notin \mathcal{N}_i^{k,Z}, j \in \mathcal{N}_i^{k,X}} (r_Z(i,j) - k) \right\}$$

where $k_1$ and $k_2$ are the bounds of the number of nearest neighbors.

(6) $d$-**RMSE** (Root Mean Square Error) measures to what extent the two distributions of **distances** coincide:

$$d - RMSE = \sqrt{\frac{1}{N^2} \sum_{i=1}^{N} \sum_{j=1}^{N} (d_X(i,j) - d_Z(i,j))^2}$$

(7) **LGD** (Locally Geometric Distortion) measures how much corresponding distances between neighboring points differ in two metric spaces and is the primary metric for isometry, defined as:

$$LGD = \sum_{k=k_1}^{k_2} \sqrt{\sum_i^M \frac{\sum_{j \in \mathcal{N}_i^{k,(l)}}(d_l(i,j) - d_{l'}(i,j))^2}{(k_2 - k_1 + 1)^2 M(\#\mathcal{N}_i)}}.$$

where $k_1$ and $k_2$ are the lower and upper bounds of the $k$-NN.

(8) **CRA** (Cluster Rank Accuracy) measures the changes in *ranks* of cluster centers from the input space $X$ and to the latent space $Z$:

$$CRA = \frac{\sum_{i=1}^C \sum_{j=1}^C \mathbf{1}(r_X(v_i^X, v_j^X) = r_Z(v_i^Z, v_j^Z))}{C^2}$$

where $C$ is the number of clusters, $v_j^X$ is the cluster center of the $j$th cluster in the input space $X$, $v_j^Z$ is the cluster center of the $j$th cluster in the latent space $Z$, $r_X(v_i^X, v_j^X)$ denotes the rank of the closeness (in terms of Euclidean distance) of $v_i^X$ to $v_j^X$ in space $X$ in the input space $X$, and $r_Z(v_i^Z, v_j^Z)$ denotes the rank of the closeness (in terms of Euclidean distance) of $v_i^Z$ to $v_j^Z$ in space $Z$.

## A.5 VISUALIZATION IN GENERALIZABILITY

The visualization results on the testing samples are shown in Fig A2; even for testing samples, our method still shows distinguishable inter-cluster discriminability, while all the other methods without exception coupled different clusters together.

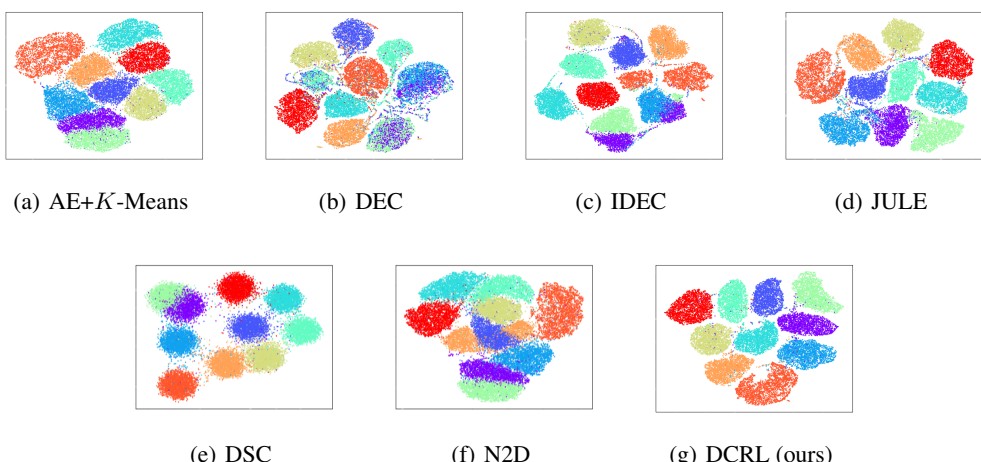

(a) AE+$K$-Means     (b) DEC     (c) IDEC     (d) JULE

(e) DSC     (f) N2D     (g) DCRL (ours)

Figure A2: The visualization of the obtained embeddings on the testing samples to show the generalization performance of different algorithms on MNIST-full dateset.

## A.6 VISUALIZATION IN DIFFERENT STAGES

The embedding visualization of the latent space during the training process is visualized in Fig A3 for depicting how both clustering and structure-preserving is achieved. We can see that the different clusters initialized by pretrained autoencoder are closely adjacent. In the early stage of training, with clustering loss $L_{cluster}$ and global ranking loss $L_{rank}$, different manifolds are separated from each other, each manifold loses its local structure, and all of them degenerate into spheres. As

the training progresses, the weight $\alpha$ for $L_{cluster}$ gradually decreases, while the weight $\beta$ for $L_{iso}$ increases and **the optimization is gradually focused from global to local**, with each manifold gradually recovering its original geometric structure from the sphere. Moreover, since our local isometry loss $L_{iso}$ is constrained within each manifold, the preservation of local structure will not disrupt the global ranking. Finally, we obtain representations in which cluster boundaries are clearly distinguished, and local and global structures are perfectly preserved. This shows that clustering-oriented loss does deteriorate the local geometric structure of the latent space, and designed $L_{LIS}$ helps to recover it.

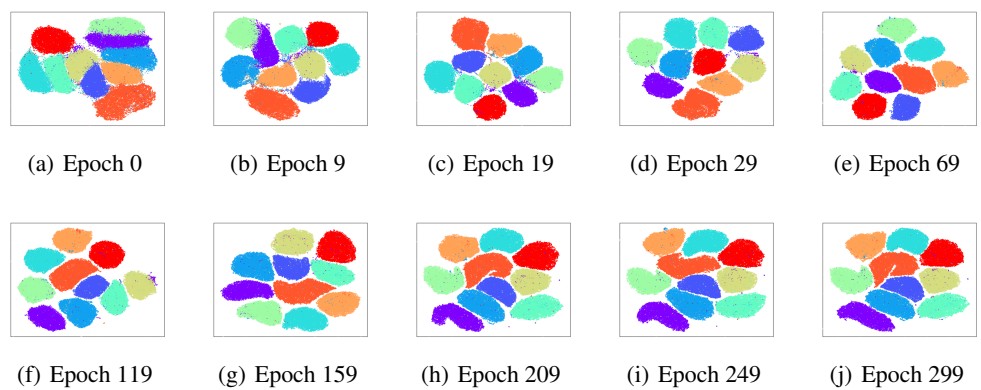

| (a) Epoch 0 | (b) Epoch 9 | (c) Epoch 19 | (d) Epoch 29 | (e) Epoch 69 |
| (f) Epoch 119 | (g) Epoch 159 | (h) Epoch 209 | (i) Epoch 249 | (j) Epoch 299 |

Figure A3: Clustering visualization at different stages of training on MNIST-full dateset.

### A.7 Exploration on the assumed cluster number $C$

Taking MNIST-test dataset as an example, we present the embedding visualization with assumed number of clusters $C$ being 10, 11, and 12, respectively. We find that when $C$ is larger than the number of true clusters (10), data originally belonging to the same cluster will be split, e.g., a cluster is split into two, but the different categories of data still hold clear boundaries and are not mixed together, somewhat similar to hierarchical clustering.

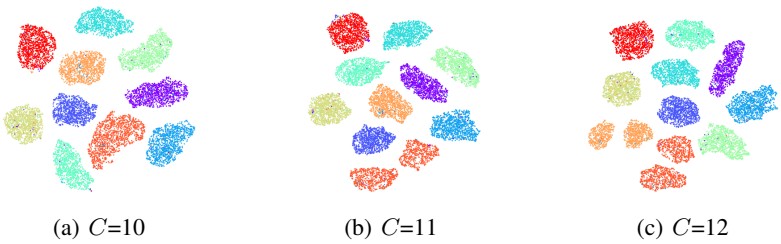

| (a) $C$=10 | (b) $C$=11 | (c) $C$=12 |

Figure A4: Clustering visualization with different assumed cluster number $C$ on MNIST-test dateset.

A.8 QUANTITATIVE EVALUATION OF REPRESENTATION LEARNING

Our method is compared with the other five methods in six evaluation metrics on six datasets. The complete results in Tab A2 demonstrate the superiority of our method, especially on metrics RRE, Trust, Cont, and CRA. As shown in Tab A2, DCRL outperforms all other methods, especially in the CRA metric, which is not only the best on all datasets but also reaches 1.0. This means that the "rank" between different manifolds in the latent space is completely preserved and undamaged, which proves the effectiveness of our global ranking loss $L_{rank}$.

Table A2: Representation learning performance of different algorithms on five datasets.

| Datasets | Algorithms | RRE | Trust | Cont | $d$-RMSE | LGD | CRA |
|---|---|---|---|---|---|---|---|
| MNIST-full | DEC | 0.09988 | 0.84499 | 0.94805 | 44.8535 | 4.37986 | 0.28 |
| | IDEC | 0.00984 | 0.99821 | 0.97936 | 24.5803 | 1.71484 | 0.33 |
| | JULE | 0.02657 | 0.93675 | 0.98321 | 28.3412 | 2.12955 | 0.27 |
| | DSC | 0.09785 | 0.87315 | 0.92508 | 6.98098 | 1.19886 | 0.23 |
| | N2D | 0.01002 | 0.99243 | 0.98466 | 5.7162 | 0.69946 | 0.21 |
| | DCRL | **0.00567** | **0.99978** | **0.98716** | **5.4986** | **0.69168** | **1.0** |
| MNIST-test | DEC | 0.12800 | 0.81841 | 0.91767 | 14.6113 | 2.29499 | 0.19 |
| | IDEC | 0.01505 | 0.99403 | 0.97082 | 7.4599 | 1.08350 | 0.38 |
| | JULE | 0.04122 | 0.92971 | 0.97208 | 9.4768 | 1.17176 | 0.42 |
| | DSC | 0.10728 | 0.85498 | 0.92254 | 7.1689 | 1.19239 | 0.26 |
| | N2D | 0.01565 | 0.98764 | 0.97572 | **5.0120** | 0.97454 | 0.33 |
| | DCRL | **0.01090** | **0.99811** | **0.97612** | 5.8000 | **0.93394** | **1.0** |
| USPS | DEC | 0.07911 | 0.88871 | 0.94628 | 16.4355 | 1.77848 | 0.31 |
| | IDEC | 0.01043 | 0.99726 | 0.97960 | 13.0573 | 1.11689 | 0.30 |
| | JULE | 0.02972 | 0.98763 | 0.98810 | 14.6324 | 1.43426 | 0.33 |
| | DSC | 0.06319 | 0.9151 | 0.93988 | 8.4412 | 1.02131 | 0.27 |
| | N2D | 0.01337 | 0.98769 | 0.98135 | 8.1961 | 0.54967 | 0.37 |
| | DCRL | **0.00577** | **0.99979** | **0.98701** | **6.4980** | **0.53180** | **1.0** |
| Fasion-MNIST | DEC | 0.04787 | 0.93896 | 0.95450 | 39.3274 | 3.87731 | 0.37 |
| | IDEC | 0.01089 | 0.99683 | 0.97797 | 25.4024 | 1.91385 | 0.27 |
| | JULE | 0.03013 | 0.97732 | 0.97923 | 15.2213 | 1.43642 | 0.43 |
| | DSC | 0.05168 | 0.95013 | 0.96121 | 17.2201 | 1.42091 | 0.36 |
| | N2D | 0.00894 | 0.99062 | 0.98054 | 14.49079 | **1.28180** | 0.26 |
| | DCRL | **0.00836** | **0.99868** | **0.98203** | **13.3788** | 1.33893 | **1.0** |
| REUTERS-10K | DEC | 0.26192 | 0.65518 | 0.80477 | 40.4671 | 4.00423 | 0.63 |
| | IDEC | 0.05981 | 0.95840 | 0.90550 | 43.9556 | 2.01365 | 0.75 |
| | JULE | - | - | - | - | - | - |
| | DSC | - | - | - | - | - | - |
| | N2D | 0.03827 | 0.97385 | 0.93412 | 36.1042 | **1.69013** | 0.31 |
| | DCRL (ours) | **0.03206** | **0.98380** | **0.93802** | **34.5478** | 2.72096 | **1.0** |
| HAR | DEC | 0.09060 | 0.89097 | 0.91766 | 10.0222 | 1.58691 | 0.30 |
| | IDEC | 0.01031 | 0.99433 | 0.98132 | 9.9155 | 0.93736 | 0.39 |
| | JULE | - | - | - | - | - | - |
| | DSC | - | - | - | - | - | - |
| | N2D | 0.00841 | 0.99281 | 0.97695 | **8.2326** | 0.64296 | 0.33 |
| | DCRL (ours) | **0.00665** | **0.99895** | **0.98634** | 15.2876 | **0.46189** | **1.0** |

A.9 STATISTICAL ANALYSIS

The statistical analysis is presented to show the extent to which local and global structure is preserved from the input space to the latent space. Taking MNIST-full as an example, the statistical analysis of the global rank-preservation is shown in Fig A5 (a)-(f). For the $i$-th cluster, if the rank (in terms of Euclidean distance) between it and the $j$-th cluster is preserved from input space to latent space, then the grid in the $i$-th row and $j$-th column is marked as blue, otherwise yellow. As shown in the figure, only our method can fully preserve the global rank between different clusters, while all other methods fail.

Finally, we perform statistical analysis for the local isometry property of each algorithm. For each sample $x_i$ in the dataset, it forms a number of point pairs with its neighborhood samples $\{(x_i, x_j)|i = 1, 2, ..., N; x_j \in \mathcal{N}_i^X\}$. We compute the difference in the distance of these point pairs from the input space to the latent space $\{d_Z(x_i, x_j) - d_X(x_i, x_j)|i = 1, 2, ..., N; x_j \in \mathcal{N}_i\}$, and plot it as a histogram. As shown in Fig A5 (g), the curves of DCRL are distributed on both sides of the 0 value, with maximum peak height and minimum peak-bottom width, respectively, which indicates that DCRL achieves the best local isometry. Although IDEC claims that they can preserve the local structure well, there is still a big gap between their results and ours.

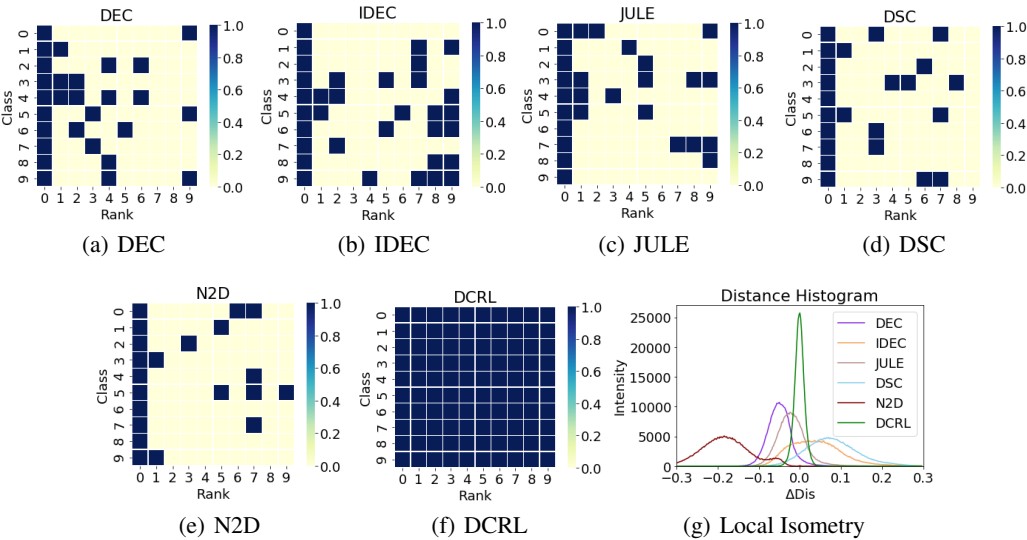

Figure A5: Statistical analysis of different algorithms to compare the capability of global and local structure preservation from the input space to the latent space.

A.10 QUANTITATIVE EVALUATION OF DOWNSTREAM TASKS

Numerous deep clustering algorithms have recently claimed to obtain meaningful representations, however, they do not analyze and experiment with the so-called "meaningful" ones. Therefore, we are interested in whether these proposed methods can indeed learn representations that are useful for downstream tasks. Four different classifiers, including a linear classifier (Logistic Regression; LR), two nonlinear classifiers (MLP, SVM), and a tree-based classifier (Random Forest Classifier; RFC) are used as downstream tasks, all of which use default parameters and default implementations in sklearn (Pedregosa et al., 2011) for a fair comparison. The learned representations are frozen and used as input for training. The classification accuracy evaluated on the test set serves as a metric to evaluate the effectiveness of learned representations. In Tab A3, DCRL outperformed the other methods overall on all six datasets, with MLP, RFC, and LR as downstream tasks. Additionally, we surprisingly find that with MLP and RFC as downstream tasks, all methods other than DCRL do not even match the accuracy of AE on the MNIST-full dataset. Notably, DEC and IDEC show a sharp deterioration in performance on downstream tasks, falling short of even the simplest AEs, again showing that clustering-oriented loss can disrupt the geometry of the data.

Table A3: Performance of different algorithms in downstream tasks.

| Datasets | Algorithms | MLP | RFC | SVM | LR |
|---|---|---|---|---|---|
| MNIST-full | AE | 0.9746 | 0.9652 | 0.9859 | 0.9565 |
| | DEC | 0.8647 | 0.8706 | 0.8707 | 0.8566 |
| | IDEC | 0.9797 | 0.9737 | 0.9852 | 0.9650 |
| | JULE | 0.9802 | 0.9825 | 0.9787 | 0.9743 |
| | DSC | 0.9622 | 0.9501 | 0.9837 | 0.9752 |
| | N2D | 0.9796 | 0.9803 | 0.9799 | 0.9792 |
| | DCRL | **0.9851** | **0.9874** | **0.9869** | **0.9841** |
| MNIST-test | AE | 0.9415 | 0.9420 | 0.9745 | 0.9495 |
| | DEC | 0.8525 | 0.8605 | 0.8725 | 0.8685 |
| | IDEC | 0.9740 | 0.9725 | 0.9845 | 0.9655 |
| | JULE | 0.9775 | 0.9845 | 0.9800 | 0.9825 |
| | DSC | 0.9535 | 0.9740 | 0.9825 | 0.9795 |
| | N2D | 0.9715 | 0.9760 | 0.9725 | 0.9725 |
| | DCRL | **0.9855** | **0.9875** | **0.9865** | **0.9855** |
| USPS | AE | 0.9421 | 0.9469 | 0.9677 | 0.9073 |
| | DEC | 0.8289 | 0.8668 | 0.8289 | 0.8294 |
| | IDEC | 0.9482 | 0.9556 | 0.9656 | 0.9125 |
| | JULE | 0.9576 | 0.9617 | **0.9703** | 0.9476 |
| | DSC | 0.9351 | 0.9572 | 0.9612 | 0.9342 |
| | N2D | 0.9569 | 0.9569 | 0.9569 | 0.9541 |
| | DCRL | **0.9656** | **0.9651** | 0.9604 | **0.9551** |
| Fasion-MNIST | AE | 0.8613 | 0.9932 | 0.8314 | 0.7588 |
| | DEC | 0.6268 | 0.9853 | 0.6377 | 0.6245 |
| | IDEC | 0.8367 | 0.9918 | **0.8607** | 0.7514 |
| | JULE | 0.8541 | 0.9892 | 0.8566 | 0.7723 |
| | DSC | 0.8084 | 0.9823 | 0.8618 | 0.7676 |
| | N2D | 0.8412 | 0.9493 | 0.8230 | 0.7753 |
| | DCRL | **0.8642** | **0.9942** | 0.8468 | **0.7768** |
| REUTERS-10K | AE | 0.9325 | 0.9170 | 0.9375 | 0.8205 |
| | DEC | 0.7985 | 0.7880 | 0.8105 | 0.7450 |
| | IDEC | 0.9225 | 0.8930 | 0.9280 | 0.7705 |
| | JULE | - | - | - | - |
| | DSC | - | - | - | - |
| | N2D | 0.9205 | 0.9080 | 0.9240 | 0.8335 |
| | DCRL (ours) | **0.9360** | **0.9185** | **0.9390** | **0.8475** |
| HAR | AE | 0.9181 | 0.9139 | 0.9201 | 0.8849 |
| | DEC | 0.7696 | 0.7847 | 0.7628 | 0.7634 |
| | IDEC | 0.8973 | 0.9031 | 0.9041 | 0.8822 |
| | JULE | - | - | - | - |
| | DSC | - | - | - | - |
| | N2D | 0.9138 | 0.9083 | 0.9174 | 0.8799 |
| | DCRL (ours) | **0.9235** | **0.9193** | **0.9293** | **0.8996** |

## A.11 MORE ABLATION EXPERIMENTS

The results of the ablation experiments on the MNIST-full dataset have been presented in Tab 4 in Sec 4.3. Here, we provide four more sets of ablation experiments on the other four datasets. The conclusion is similar (note that the clustering performance of the model without clustering-oriented losses is very poorly, so the "best" metric numbers are not meaningful and are shown in gray color): (1) CL is very important for obtaining good clustering. (2) SL is beneficial for both clustering and representation learning. (3) Our training strategies (WC and AT) are very superior in improving metrics such as ACC, RRE, Trust, Cont, and CRA.

Table A4: Ablation study of loss items and training strategies used in DCRL.

| Datasets | Methods | ACC/NMI | RRE | Trust | Cont | $d$-RMSE | LGD | CRA |
|---|---|---|---|---|---|---|---|---|
| MNIST-full | w/o SL | 0.976/0.939 | 0.0093 | 0.9967 | 0.9816 | 24.589 | 1.6747 | 0.32 |
| | w/o CL | 0.814/0.736 | 0.0004 | 0.9998 | 0.9990 | 7.458 | 0.0487 | 1.00 |
| | w/o WC | 0.977/0.943 | 0.0065 | 0.9987 | 0.9860 | 5.576 | 0.6968 | 0.98 |
| | w/o AT | 0.978/0.944 | 0.0069 | 0.9986 | 0.9851 | 5.617 | 0.7037 | 0.96 |
| | full model | **0.980/0.946** | **0.0056** | **0.9997** | **0.9871** | **5.498** | **0.6916** | **1.00** |
| MNIST-test | w/o SL | **0.973/0.932** | 0.0146 | 0.9928 | 0.9727 | 7.701 | 1.0578 | 0.31 |
| | w/o CL | 0.773/0.747 | 0.0020 | 0.9994 | 0.9954 | 7.229 | 0.0809 | 1.00 |
| | w/o WC | 0.956/0.904 | 0.0132 | 0.9955 | 0.9735 | **5.470** | 0.9364 | **1.00** |
| | w/o AT | 0.970/0.929 | 0.0118 | 0.9974 | 0.9747 | 5.567 | 0.9404 | **1.00** |
| | full model | 0.972/0.930 | **0.0109** | **0.9981** | **0.9761** | 5.800 | **0.9339** | **1.00** |
| USPS | w/o SL | 0.958/0.902 | 0.0095 | 0.9967 | 0.9812 | 14.609 | 0.9847 | 0.29 |
| | w/o CL | 0.664/0.658 | 0.0020 | 0.9996 | 0.9952 | 2.934 | 0.0687 | 1.0 |
| | w/o WC | 0.956/0.896 | 0.0060 | 0.9991 | 0.9868 | 6.572 | 0.5335 | **1.00** |
| | w/o AT | 0.947/0.885 | 0.0080 | 0.9979 | 0.9833 | **5.960** | **0.4967** | **1.00** |
| | full model | **0.960/0.902** | **0.0057** | **0.9997** | **0.9870** | 6.498 | 0.5318 | **1.00** |
| Fasion-MNIST | w/o SL | 0.706/0.682 | 0.0108 | 0.9964 | 0.9781 | 25.954 | 1.8936 | 0.30 |
| | w/o CL | 0.576/0.569 | 0.0004 | 0.9994 | 0.9995 | 7.654 | 0.0523 | 1.00 |
| | w/o WC | 0.702/0.695 | 0.0084 | 0.9972 | 0.9814 | **13.238** | 1.3474 | **1.00** |
| | w/o AT | 0.708/0.694 | 0.0097 | 0.9975 | 0.9798 | 13.354 | 1.3611 | **1.00** |
| | full model | **0.710/0.685** | **0.0083** | **0.9986** | **0.9820** | 13.378 | **1.3389** | **1.00** |
| REUTERS-10K | w/o SL | 0.819/0.564 | 0.0529 | 0.9610 | 0.9185 | 44.481 | 1.9090 | 0.38 |
| | w/o CL | 0.542/0.279 | 0.0277 | 0.9868 | 0.9456 | 37.018 | 2.2294 | 1.00 |
| | w/o WC | 0.830/0.583 | 0.0420 | 0.9667 | 0.9361 | 35.302 | 2.8286 | **1.00** |
| | w/o AT | 0.825/0.563 | 0.0440 | 0.9650 | 0.9330 | 39.275 | 2.9146 | **1.00** |
| | full model | **0.836/0.590** | **0.0320** | **0.9838** | **0.9380** | **34.547** | **2.7209** | **1.00** |
| HAR | w/o SL | 0.835/0.746 | 0.0116 | 0.9944 | 0.9792 | **8.168** | 0.8882 | 0.33 |
| | w/o CL | 0.744/0.615 | 0.0024 | 0.9986 | 0.9948 | 15.060 | 0.2193 | 1.00 |
| | w/o WC | 0.786/0.701 | 0.0130 | 0.9950 | 0.9756 | 15.398 | 0.6171 | **1.00** |
| | w/o AT | 0.834/0.745 | 0.0089 | 0.9965 | 0.9835 | 15.726 | 0.4734 | **1.00** |
| | full model | **0.845/0.758** | **0.0066** | **0.9989** | **0.9863** | 15.287 | **0.4618** | **1.00** |

## A.12 PARAMETER SENSITIVITY

We also evaluated the sensitivity of parameters $k$ and $\kappa$ on the MNIST-test dataset and the results are shown in Tab A5. The parameters $k$ and $\kappa$ are found to have little effect on the clustering performance (ACC/NMI), and some combinations of $k$ and $\kappa$ even produce better clustering performance than the metrics reported in the main paper. However, the effect of $k$ and $\kappa$ on representation learning is more pronounced, and different combinations of $k$ and $\kappa$ may increase or decrease performance. In general, this paper focuses on the design of the algorithm itself and has not performed the parameter search to find the best performance.

Table A5: Parameter Sensitivity with different parameters $k$ and $\kappa$ on the MNIST-test dataset.

| Parameters | ACC/NMI | RRE | Trust | Cont | $d$-RMSE | LGD | CRA |
|---|---|---|---|---|---|---|---|
| $k$=1, $\kappa$=3 | **0.975/0.936** | 0.0125 | 0.9944 | 0.9756 | 5.757 | **0.8868** | 1.00 |
| $k$=3, $\kappa$=3 | 0.973/0.931 | 0.0114 | 0.9970 | 0.9757 | 5.805 | 0.9207 | 1.00 |
| $k$=5, $\kappa$=3 | 0.972/0.930 | 0.0109 | 0.9981 | 0.9761 | 5.800 | 0.9339 | 1.00 |
| $k$=8, $\kappa$=3 | 0.972/0.929 | **0.0104** | 0.9989 | **0.9765** | 5.810 | 0.9476 | 1.00 |
| $k$=10, $\kappa$=3 | 0.972/0.929 | 0.0105 | **0.9990** | 0.9764 | **5.704** | 0.9487 | 1.00 |
| $k$=5, $\kappa$=1 | 0.967/0.912 | **0.0068** | **0.9993** | **0.9845** | **5.409** | **0.2524** | 1.00 |
| $k$=5, $\kappa$=3 | **0.972/0.930** | 0.0109 | 0.9981 | 0.9761 | 5.800 | 0.9339 | 1.00 |
| $k$=5, $\kappa$=5 | 0.972/0.929 | 0.0146 | 0.9964 | 0.9691 | 15.0653 | 1.5719 | 1.00 |
| $k$=5, $\kappa$=8 | 0.972/0.929 | 0.0190 | 0.9943 | 0.9615 | 29.4607 | 2.5410 | 1.00 |
| $k$=5, $\kappa$=10 | 0.972/0.929 | 0.0195 | 0.9951 | 0.9597 | 37.7661 | 3.1434 | 1.00 |

