# OpenReview forum: "Deep Clustering and Representation Learning that Preserves Geometric Structures"
_ICLR.cc/2021/Conference — Reject_

### Official Review · AnonReviewer2 · 2020-10-26
**Ok but not good enough - rejection**

**Rating:** 4
**Confidence:** 4

**Review:**

This paper proposes a method which can simultaneously perform clustering and represent learning with local and global structure preservation. The method imposes clustering-oriented and structure-oriented in order to optimize the latent space. The experimental results show its effectiveness. However, here are some issues.
(1) This paper is not innovative enough. From the perspective of the entire clustering process, the proposed clustering method essentially makes the intra-cluster distance as small as possible and the inter-cluster distance as large as possible. There is no significant innovation in the idea of clustering.
(2) In the initialization of the algorithm, first use t-SNE to transform the latent space Z into two dimensions, and then run K-Means to get the label of each data point. In this step, why do you need to perform dimension conversion first instead of directly performing K-Means clustering on the original data?
(3) In this method, the author uses K-Means for initialization. However, K-Means is not a stable clustering algorithm. Using it to initialize the label, are the results stable? Has the author analyzed the effect of different label initialization methods on the results?
(4) The data sets used in the experiment are too single. In the adopted data sets, three are all about MNIST, and there is little difference between MNIST-test and MNIST-full. Moreover, similar to MNIST, USPS is also a handwritten digital image data set. The data sets used by the author are repetitive, and it is difficult to reflect the experimental performance of the method.
(5) From the quantitative evaluation results of downstream tasks, compared with the classic classification algorithms, the representation learning obtained by the proposed method for classification does not show great advantages in classification accuracy.
Overall, I think this paper is not ready yet.

---

> ### Author Response · Authors · 2020-11-24
> **Response to reviewer 4 (#1)**
>
> Thanks for your insightful reviews! We address your concerns and questions as follows:
> - Q1: This paper is not innovative enough. From the perspective of the entire clustering process, the proposed clustering method essentially makes the intra-cluster distance as small as possible, and the inter-cluster distance as large as possible. There is no significant innovation in the idea of clustering.
> - Answer1:
> -- **A.** Thank you for your comment. Making intra-cluster distances as small as possible and inter-cluster distances as large as possible is the goal of all clustering algorithms, the question is how to go about achieving this goal? This paper is not intended to innovate on the idea, concept of clustering, but rather to propose a solution based on learnable cluster centers that constantly updates the label assignment in an iterative manner, and ultimately enables clustering.
> -- **B.** First, we want to restate the relationship between geometric structure preservation and multi-manifold representation learning to avoid any possible misunderstanding. What we actually addressing here is **unsupervised multi-manifold learning**, but due to page limitations and to avoid excessive repetition, we sometimes simplify it as **representational learning**. （In the revised manuscript, all relevant statements have been modified.) Unlike general representation learning, based on the manifold assumption, manifold representations should satisfy the following two main points: **Point (1)** preserving the local geometric structure within each manifold and **Point (2)** ensuring the discriminability between different manifolds. General representational learning concerns only **Point (2)**, which is consistent with the goal of clustering, but **Point (1)** sometimes contradicts with clustering (the contraction has been analyzed in Section 3.3 in the paper). In this paper, we propose an elegant training strategy to mitigate this contradiction so that clustering helps both Point (1) and Point (2).
> -- **C.** In the end, we achieved the following goals related to clustering, geometric structures, and manifold representations: **(1) Clustering helps Point (2)** - ensure the discriminability between different manifolds: clear inter-cluster boundaries in Fig. 4, statistical analysis of global structure preservation in Fig. A5(a)-(e); **(2) Local structure can be preserved even in the presence of clustering**: better performance for representation learning and downstream tasks (Table A2 and Table A3), the embedding visualizations of clustering process in Fig. A3, and various cluster shapes (not all deteriorated to spherical) in Fig. 4 ; **(3) Geometric structure preservation helps clustering**: better clustering performance in Table 1 and excellent generalizability (Table 2 and Fig. A2). Furthermore, in the ablation experiments in Table A4, we also show the importance of structure-oriented losses to improve clustering and multi-manifold representation learning performance.
> -- **D.** The above analyses and explanations have been added in the corresponding parts in the revised manuscript, especially in the introduction section, where we have added more of our motivations and objectives.
>
> ---
> - Q2:  Why do you need to perform dimension conversion first instead of directly performing K-Means clustering on the original data?
> - Answer2: A. Thanks for your comment. Due to the non-Euclidean property of the high dimensional data therein, performing the Euclidean distance-based K-means algorithm directly on the raw data yields poor results. Better performance can be achieved by performing dimensionality reduction on the data first, followed by clustering in a lower-dimensional space, which has been confirmed by the experiments in the N2D paper [1].
>
> ---
> - Q3: K-Means is not a stable clustering algorithm. Using it to initialize the label, are the results stable? Has the author analyzed the effect of different label initialization methods on the results?
> - Answer3: Due to the non-Euclidean property of high-dimensional data, the results of K-means are unstable in high-dimensional space with good results under some random seeds and poor results under others, which we believe is because the higher the space dimensionality, the less reliable the Euclidean metric K-Means based are.  However, we find that the results of K-Means are relatively stable in 2D space, where the direct use of the Euclidean metric is more reliable. We experimented with it and found that it is not sensitive to various initialization methods, so we went straight to the simplest K-means. **Another important reason is that K-means is an algorithm based on the center of mass, which coincides with our notion of learnable cluster centers**.  As mentioned in the paper, our cluster centers are **learnable**, each epoch can be updated, and a good initialization method is just to find a good starting point for optimization, which does not necessarily mean better performance.

---

> ### Author Response · Authors · 2020-11-24
> **Response to reviewer 4 (#2)**
>
> ---
> - Q4: The data sets used in the experiment are too single. In the adopted data sets, three are all about MNIST, and there is little difference between MNIST-test and MNIST-full. Moreover, similar to MNIST, USPS is also a handwritten digital image data set. The data sets used by the author are repetitive, and it is difficult to reflect the experimental performance of the method.
> - Answer4:
> -- **A.** Thanks for the comments. Most current deep clustering algorithms use both MNIST and USPS datasets for testing, even though the two datasets are very similar. For the MNIST data, some use the full MNIST data (MNIST-full), others use only the MNIST-test part (MNIST-test). For a fair comparison, MNIST-full and MNIST-test are usually compared **separately** in most work, such as [1,2,3].
> -- **B.** Additionally, we have added a comparison with ASPD-DA [4], the current SOTA method for clustering. Note that in the ASPD-DA paper, they also evaluated performance on just only four datasets (MNIST-test, MNIST-full, USPS, and Fashion-MNIST), which are similar to our settings. While ASPC-DA achieves better performance on some datasets, its performance gains do not come directly from clustering, but from sophisticated modules such as data augmentation and self-paced learning. Once these modules are removed, there is a very large degradation in its performance. For example, with data augmentation removed, ASPC-DA achieves less competitive performance, e.g., an accuracy of 0.924 (vs 0.988) on MNIST-test, 0.785 (vs 0.973) on MNIST-full and 0.688 (vs 0.982) on USPS. In addition, although ASPC-DA is based on the MLP architecture, its image-based data augmentation methods cannot be directly applied to vector data, which explains why ASPC has not any performance advantage on vector-based REUTERS-10K and HAR dataset (even compared to baseline algorithms DEC and IDEC).
> -- **C**. Thank you again for your suggestion, and to highlight our strengths, especially in **vector-based** datasets, we have experimented with another vector-based dataset HAR in the revised manuscript.
>
> ---
> - Q5: From the quantitative evaluation results of downstream tasks, compared with the classic classification algorithms, the representation learning obtained by the proposed method for classification does not show great advantages in classification accuracy.
> - Answer5:
> -- **A.** Thank you for your comment. Since the advantage of evaluating learned representation through downstream tasks is not significant, we have removed this experiment from the main paper and placed it in the Appendix 10.
> -- **B.** As Table A3 shows, although the advantages are not clear, we still achieved the best on almost all datasets. Besides, the downstream task is only one of the many evaluation tasks in this paper; other evaluations include clustering performance (Table 1), clustering visualization (Fig. 4), and clustering generalizability (Table 2 and Fig. A2). Experiments related to representation learning include embedding visualization during clustering process (Fig. A3), representation learning performance (Table A2), and statistical analysis of geometric preservation (Fig. A5).
>
> ---
> [1] Ryan McConville, Raul Santos-Rodriguez, Robert J Piechocki, and Ian Craddock. N2d:(not too) deep clustering via clustering the local manifold of an autoencoded embedding. arXiv preprint arXiv:1908.05968, 2019.
>
> [2] Yang J, Parikh D, Batra D. Joint unsupervised learning of deep representations and image clusters[C]//Proceedings of the IEEE Conference on Computer Vision and Pattern Recognition. 2016: 5147-5156.
>
> [3] Yang X, Deng C, Zheng F, et al. Deep spectral clustering using dual autoencoder network[C]//Proceedings of the IEEE Conference on Computer Vision and Pattern Recognition. 2019: 4066-4075.
>
> [4] Xifeng Guo, Xinwang Liu, En Zhu, Xinzhong Zhu, Miaomiao Li, Xin Xu, and Jianping Yin. Adaptive self-paced deep clustering with data augmentation.IEEE Transactions on Knowledge and Data Engineering, 2019.

---

### Official Review · AnonReviewer1 · 2020-10-27
**Good work but the main claim needs a more specific illustration**

**Rating:** 6
**Confidence:** 4

**Review:**

[Summary]

In this paper, the authors proposes a deep clustering model to enable the clustering and representation learning to favor each other via preserving the geometric structure of data. The proposed DCRL framework integrates an isometric loss for local intra-manifold structure and a ranking loss for global inter-manifold structure. The authors evaluate the proposed framework on five datasets and the experimental results show that the proposed framework brings certain improvements over the baseline approaches.
Generally speaking, it is a good work with good results. Considering there are some weaknesses to be solved, I’d like to vote for a rating of 6.

[Pros]
+ The paper is well organized and easy to follow.
+ The idea of preserving global and local structure is reasonable and seems to be effective.
+ The description of experimental setting is sufficient, should be easy to reproduce. The authors also conduct comprehensive experiments with good results achieved.

[Cons]
- The proposed intra-manifold isometry loss and inter-manifold ranking loss seem to be modifications of $L_{LIS}$ proposed in [1] so the framework is interesting but not surprisingly new to me.
- Though the authors claim that ‘the clustering-oriented loss may deteriorate the geometric structure of the latent space’ at the very beginning of this paper, this claim is not supported by any proof until it reaches the end of the experiment section. It would be better to illustrate this phenomenon by some figures or case studies in the introduction to help the readers understand this. Moreover, the relationship between geometric structure and the clustering performance is also recommended to be analyzed to show the necessity of preserving the local data structure.
- In this paper, the meaningfulness of learned representations is tested on downstream classification tasks. However, it seems not very reasonable to adopt such evaluation, since intuitively the representation learning should focus on improving the clustering performance and it seems not very necessary to take the downstream task into account for the clustering models (at least for me it is not very practical to use representations learned by deep clustering models to do some classifications).
- Experiments are only conducted only with MLP-AE backbones. It is not clear whether the data structure will be preserved for ConvAEs which is also commonly used in real-world applications.
- There are some minor issues, e.g., 'learining' in page 2 and page 17 should be 'learning'. In Fig.4, ‘AE+K-mean’ should be ‘AE-K-means’.
[1] Stan Z Li, Zelin Zhang, and Lirong Wu. Markov-lipschitz deep learning. arXiv preprint arXiv:2006.08256, 2020.

[Questions]
1. Could you please provide some explanation for why the clustering-oriented loss may deteriorate the geometric structure of the latent space and why preserving the structure is necessary for the clustering?
2. Will the data structure be preserved for ConvAEs backbones?
3. How do the hyperparameters $k$ and $scale$ affect the performance?
4. How to set the the range of hyperparameters $\alpha, \beta$ in practice?

---

> ### Author Response · Authors · 2020-11-24
> **Response to reviewer 3 (#1)**
>
> Thanks for your insightful reviews! We address your concerns and questions as follows:
> - Q1: The proposed intra-manifold isometry loss and inter-manifold ranking loss seem to be modifications of  proposed in [1] so the framework is interesting but not surprisingly new to me.
> - Answer1: The LIS loss is a local distance-preserving loss that is widely used in algorithms for multi-manifold learning. However, other approaches usually only apply it to all **N** sample points, dealing with $N×N$ point-to-point relationships, which is computationally very large and difficult to optimize. Instead, we take the label assignment from each epoch into account, making this constraint **confined within each manifold**, which is simpler and easier to optimize.
>
> ---
> - Q2: It would be better to illustrate this phenomenon by some figures or case studies in the introduction to help the readers understand this.
> - Answer2: Thank you for your suggestions. We have re-layout the manuscript provide experiments and analyses related to the deterioration of geometry structure in the introduction section.
>
> ---
> - Q3: Could you please provide some explanation for why the clustering-oriented loss may deteriorate the geometric structure of the latent space and why preserving the structure is necessary for the clustering?
> - Answer3:
> -- **A.** Thanks for your comment. First, this claim was first made in the IDEC paper, which says：“the defined clustering loss may corrupt feature space, which leads to non-representative meaningless features and this, in turn, hurts clustering performance.” and “The autoencoder is used to learn representations in an unsupervised manner and the learned features can preserve intrinsic local structure in data.”  Despite the claim they make, they do not provide any evidence, and their solution is nothing more than preserving local structure through reconstruction loss. In this paper, on the other hand, we show through the visualization of embedding learned by DEC (Fig. 4b), the visualization of the clustering process (Fig. A3), and statistical analysis (Appendix 9) that the geometry of the latent space is indeed disrupted.  As shown in Fig. A3, in the early stage of training, with clustering-oriented loss $L_{cluster}$, each manifold loses its local structure, and all of them degenerate into spheres (geometric structure of the latent space has **deteriorated**). As the training progresses, with $L_{LIS}$ dominating, the optimization is gradually focused from global to local, with each manifold gradually recovering its original geometric structure from the sphere (the geometric structure of the latent space is **recovered**).
> -- **B.** First, we want to restate the relationship between geometric structure preservation and multi-manifold representation learning to avoid any possible misunderstanding. What we actually addressing here is **unsupervised multi-manifold learning**, but due to page limitations and to avoid excessive repetition, we sometimes simplify it as **representational learning**. （In the revised manuscript, all relevant statements have been modified.) Unlike general representation learning, based on the manifold assumption, manifold representations should satisfy the following two main points: **Point (1)** preserving the local geometric structure within each manifold and **Point (2)** ensuring the discriminability between different manifolds. General representational learning concerns only **Point (2)**, which is consistent with the goal of clustering, but **Point (1)** sometimes contradicts with clustering (the contraction has been analyzed in Section 3.3 in the paper). In this paper, we propose an elegant training strategy to mitigate this contradiction so that clustering helps both Point (1) and Point (2).
> -- **C.** In the end, we achieved the following goals related to clustering, geometric structures, and manifold representations: **(1) Clustering helps Point (2)** - ensure the discriminability between different manifolds: clear inter-cluster boundaries in Fig. 4, statistical analysis of global structure preservation in Fig. A5(a)-(e); **(2)** **Local structure can be preserved even in the presence of clustering**: better performance for representation learning and downstream tasks (Table A2 and Table A3), the embedding visualizations of clustering process in Fig. A3, and various cluster shapes (not all deteriorated to spherical) in Fig. 4 ; **(3)** **Geometric structure preservation helps clustering**: better clustering performance in Table 1 and excellent generalizability (Table 2 and Fig. A2). Furthermore, in the ablation experiments in Table A4, we also show the importance of structure-oriented losses to improve clustering and multi-manifold representation learning performance.
> -- **D.** The above analyses and explanations have been added in the corresponding parts in the revised manuscript, especially in the introduction section, where we have added more of our motivations and objectives.

---

> ### Author Response · Authors · 2020-11-24
> **Response to reviewer 3 (#2)**
>
> - Q4: In this paper, the meaningfulness of learned representations is tested on downstream classification tasks. However, it seems not very reasonable to adopt such evaluation, since intuitively the representation learning should focus on improving the clustering performance and it seems not very necessary to take the downstream task into account for the clustering models.
> - Answer4: Thanks for your comment. Although numerous previous work has claimed that they brought clustering and representation learning into a unified framework,  they all,  unfortunately,  lack an analysis of the effectiveness of the learned representations. In this paper, **from the perspective of manifold representation learning**, we measure the "meaningful" of the learned representation from both quantitative metrics comparison and downstream task. However, **from a purely clustering perspective**, we acknowledge that downstream tasks are not a necessary factor to be taken into account. Therefore, we have removed this experiment from the main paper and placed it in the Appendix 10.
>
> ---
> - Q5: Experiments are only conducted only with MLP-AE backbones. It is not clear whether the data structure will be preserved for ConvAEs which is also commonly used in real-world applications.
> - Answer5: Thank you for your suggestion, we are currently using the MLP architecture for this version and will extend it to ConvAE in the future, which has been added in the revised manuscripts. Deep clustering and visual self-supervised representation learning (SSL) are two different areas of research, although sometimes they both evaluate performance in terms of accuracy. SSL typically uses more powerful **Convolutional Neural Networks**, applicable only to image data, and uses techniques such as contrastive learning [2,3], data augmentation [2,3], and clustering [4,5,6] for better performance on large-scale datasets such as ImageNet. Deep clustering, on the other hand, uses a **generic MLP architecture** for both image and vector data, so it is difficult to scale directly to large datasets without considering those complex techniques. In this paper, instead of using the CNN architecture following the settings as unsupervised representation learning, we use the MLP architecture to design general clustering algorithms ( **applied to vector data, not just image data**) in line with the approach DEC, IDEC, etc.. We have added more content in the related work section to distinguish our work from visual self-supervised representation learning, which explains why we are currently using the MLP architecture rather than ConAE.
>
> ---
> - Q6: There are some minor issues, e.g., 'learining' in page 2 and page 17 should be 'learning'. In Fig.4, ‘AE+K-mean’ should be ‘AE-K-means’.
> - Answer6: Thank you for your suggestions, they have been corrected in the revised manuscript.
>
> ---
> - Q7: How do the hyperparameters  $k$ and $scale$  affect the performance? How to set the range of hyperparameters $\alpha$, $\beta$ in practice?
> - Answer7:
> -- **A.** Thank you for your suggestion. An experiment on the sensitivity of parameters  $k$ and $scale$ ($\kappa$ in the revised manuscript) has been performed and is included in **Appendix 12** and **Table A.5**. We have tested the effect of the parameters $k$ and $\kappa$ on performance and found that they have little effect on clustering performance. However, the effect of $k$ and $\kappa$ on representation learning performance is more significant, and different combinations of $k$ and $\kappa$ may increase or decrease the performance. Overall, this paper focuses on the design of the algorithm itself and does not perform parameters search to find the optimal performance.
> -- **B.** Thank you for your comment. Although a continuous strategy is applied to $\alpha$ and $\beta$ to make different constraints dominate at different stages (this is analyzed in detail in Section 3.3), the starting and ending number of weight graduality is 1.0 (rather than some strange number). We did not make other, more complex settings for $\alpha$ and $\beta$; we used the same settings for the parameters $\alpha$ and $\beta$ on all the datasets we tested, and all obtain good results. **We think it may be the strategy of weight graduality that is important, rather than their specific values.**
>
> ---
> [1] Stan Z Li, Zelin Zhang, and Lirong Wu. Markov-Lipschitz deep learning. arXiv preprint arXiv:2006.08256, 2020.

---

### Official Review · AnonReviewer4 · 2020-10-29
**A new deep clustering method that better accounts for the geometric structure of the data, demonstrated by comparison with similar methods.**

**Rating:** 7
**Confidence:** 4

**Review:**

The main contributions of this work are the inclusion of some new optimization objectives into the standard deep clustering formulations.
The cluster orientated loss seems to come from DEC but as the authors point out this causes issues that affect the quality of clusters.
Thus to preserve the manifolds within the embedding the propose new objectives in addition to this. One objective tries to ensure representation clusters are internally coherent with the original manifold Another objective tries to ensure global structure is preserved by optimizing directly the centroids of the learned representation

There are a number of strengths to this work.

The idea of including the geometric structure is well founded and their approach appears to account for this well. Unlike the typical work in this area, the authors choose to evaluate their representations on downstream tasks, and not just clustering, included. This really demonstrates the benefits (and limitations) outside of the typical clustering focus. I also liked the inclusion of the ablation study which shows benefit of Weight Graduality (WG) and Alternating Training (AT). They also perform sufficient experimentation to validate their approach.

One concern w.r.t. there proposed approach is that one of their objective functions only updates the centroids and thus they need another objective to align these. This seems like a rather arbitrary way of going about this.

The authors also state that "It is worth noting that all of the above losses coexist rather than independently at different stages" However, the final stage corresponds to L_LIS corresponds outside of the batching, which is a little misleading. I would not say they are not independent in this case.

I have some questions w.r.t. the evaluation..
Why is tSNE used after pretraining when the work of N2D (which you reference) shows UMAP performs better? This is more of a question about the authors choice, rather that a criticism of the work.
I also am unclear why they reduce the dimensionality to 2. This is intuitive when you are visualizing the dataset, but not necessarily for clustering. The above paper N2D (which you reference) mentions that the cluster quality lessens with 2 dimensions as opposed to higher.
Again, despite these choices, the method outperforms others, thus these are less a direct criticism of the work and more about the lack of justification these choices.

I believe the authors are missing some other relevant work in their comparisons, for e.g. [1], which reports higher performance than the authors on some datasets.

Further, like much of these works, the method is only tested on smaller datasets. Other approaches are tested on datasets such as ImageNet-10, the inclusion of more larger image datasets would strengthen this work.

[1] Guo, Xifeng, et al. "Adaptive self-paced deep clustering with data augmentation." IEEE Transactions on Knowledge and Data Engineering (2019).

---

> ### Author Response · Authors · 2020-11-24
> **Response to reviewer 2 (#1)**
>
> Thanks for your insightful reviews and we appreciate your valuable suggestions! We address your concerns and questions as follows:
> - Q1: One concern w.r.t. there proposed approach is that one of their objective functions only updates the centroids and thus they need another objective to align these. This seems like a rather arbitrary way of going about this.
> - Answer1:
> -- **A.** Thank you for your comment. Conventional methods for dealing with inter-manifold separation typically impose $L_{sep}$ in Eq. (7) on all data points from different manifolds, which involves $N×N$ point-to-point relationships. This means that each point may be subject to the push-away force from other manifolds, but at the same time, each point has to meet the isometry constraint with its neighboring points. Under these two constraints, optimization is difficult and it is easy to fall into a local optimal solution and get inaccurate results. In contrast, DCRL's $L_{rank}$ constraint does not act directly on the points, but on the clustering centers, which avoids the above problem and makes it easier to optimize. This explains why $L_{rank}$ only optimizes the cluster center and not all the sample points.
> -- **B.** Moreover, the optimization of the cluster center is not only dependent on $L_{rank}$, but is also constrained by $L_{cluster}$, which ensures that the sample points remain largely distributed around the cluster center during the optimization process and do not deviate significantly from it. Alignment loss, as an auxiliary term, aims to make this binding stronger and training more stable.
>
> ---
> - Q2: The authors also state that "It is worth noting that all of the above losses coexist rather than independently at different stages" However, the final stage corresponds to L_LIS corresponds outside of the batching, which is a little misleading. I would not say they are not independent in this case.
> - Answer2: Thanks for your suggestion, we have modified the expression in the revised manuscript to avoid any possible misunderstanding.
>
> ---
> - Q3: Why is tSNE used after pretraining when the work of N2D (which you reference) shows UMAP performs better?
> - Answer3: Thank you for your comments. In the original paper of N2D, it is not clear whether the results it provides are the best or the average. In fact, we experimented with different random seeds according to its original experimental settings, and it did not always maintain high accuracy, e.g. on the mnist-full dataset, with some random seeds, the accuracy was only about 0.870 (vs 0.979 it reported). So we guess that N2D has reported the best metrics. We tried to reproduce their results based on the hyperparameters in the N2D paper, and we found that in terms of average performance, t-SNE would even be better than UMAP if the t-SNE parameters were set properly, so unlike the approach in N2D, we used t-SNE rather than UMAP.
>
> ---
> - Q4:  I also am unclear why they reduce the dimensionality to 2. This is intuitive when you are visualizing the dataset, but not necessarily for clustering.
> - Answer4:
> -- **A.** Thank you for your comments. First, although promising results sometimes can be obtained on high-dimensional embeddings (e.g., 10D), the clustering results are not stable enough, with good results under some random seeds and poor results under other seeds, which we believe is because the higher the space dimensionality, the less reliable the Euclidean metric K-Means based are. And although clustering in 2-dimensional space is sometimes worse in terms of performance, the results are more stable.
> -- **B.** In addition, an important reason is that we use t-SNE instead of UMAP, and it would be too computationally large to use t-SNE to reduce the data to high-dimensional space (higher than 2D).
> -- **C.** Furthermore, our clustering centers are **learnable** and updated at each epoch in an iterative manner, and we think a proper initialization in 2 dimensions is sufficient for us. Thus, we have not explored this problem in much depth.

---

> ### Author Response · Authors · 2020-11-24
> **Response to reviewer 2 (#2)**
>
> - Q5: I believe the authors are missing some other relevant work in their comparisons, for e.g. [1], which reports higher performance than the authors on some datasets.
> - Answer5:
> -- **A.** Thank you for your suggestion, the reference you mentioned has been added for comparison in Table 1 in the revised manuscript. In addition, we also provide the performance of the ASPD-DA algorithm with data augmentation removed.
> -- **B.** While ASPC-DA achieves better performance on some datasets, its performance gains do not come directly from clustering, but from sophisticated modules such as data augmentation and self-paced learning. Once these modules are removed, there is a very large degradation in its performance. For example, with data augmentation removed, ASPC-DA achieves less competitive performance, e.g., an accuracy of 0.924 (vs 0.988) on MNIST-test, 0.785 (vs 0.973) on MNIST-full and 0.688 (vs 0.982) on USPS. Looking at it this way, it seems that the gain in ASPC-DA performance comes more from data augmentation?
> -- **C.** Deep clustering and visual self-supervised representation learning (SSL) are two different research areas, although sometimes they both evaluate performance in terms of accuracy. SSL typically uses more powerful Convolutional Neural Networks, applicable only to image data, and uses techniques such as contrastive learning [2,3], data augmentation [2,3], and clustering [4,5,6] for better performance on large-scale datasets such as ImageNet. Deep clustering, on the other hand, uses a generic MLP architecture for both image and vector data, so it is difficult to scale directly to large datasets without considering those sophisticated techniques. In this paper, instead of using the CNN architecture following the settings as unsupervised representation learning, we use the MLP architecture to design general clustering algorithms ( applied to vector data, not just image data) in line with the DEC, IDEC, etc.. This explains why we do not use sophisticated techniques such as data augmentation to enhance performance and extend to large datasets such as ImageNet. In a fair comparison, several of the methods we compare also focus only on the clustering algorithm itself and are not combined with other sophisticated modules. In a fair comparison, DCRL achieves the overall best performance.
> - The above analysis and explanations have been added to the revised manuscript.
>
> ---
> - Q6: Further, like much of these works, the method is only tested on smaller datasets. Other approaches are tested on datasets such as ImageNet-10, the inclusion of more larger image datasets would strengthen this work.
> - Answer6: Thank you again for your suggestion, and to highlight our strengths, especially in vector-based datasets, we have experimented with another vector-based dataset HAR in the revised manuscript. As stated in ***Answer 5***, our approach currently uses an MLP architecture that can be used to work with both vector and image data (reshaped to vector first), so it has not been tested on very large-scale datasets (in fact, most of the methods we compared have not been tested on large-scale datasets). In the future, we will consider extending it to a version of CoonAE and will try to test it on ImageNet-10.
>
> ---
> [1] Guo, Xifeng, et al. "Adaptive self-paced deep clustering with data augmentation." IEEE Transactions on Knowledge and Data Engineering (2019).
>
> [2] He K, Fan H, Wu Y, et al. Momentum contrast for unsupervised visual representation learning[C]//Proceedings of the IEEE/CVF Conference on Computer Vision and Pattern Recognition. 2020: 9729-9738.
>
> [3] Chen T, Kornblith S, Norouzi M, et al. A simple framework for contrastive learning of visual representations[J]. arXiv preprint arXiv:2002.05709, 2020.
>
> [4] Zhan X, Xie J, Liu Z, et al. Online Deep Clustering for Unsupervised Representation Learning[C]//Proceedings of the IEEE/CVF Conference on Computer Vision and Pattern Recognition. 2020: 6688-6697.
>
> [5] Xu, Ji, et al. "Invariant information clustering for unsupervised image classification and segmentation."ICCV. 2019.
>
> [6] Wouter, Van Gansbeke, et al. "Scan: Learning to classify images without labels." ECCV. 2020.

---

### Official Review · AnonReviewer3 · 2020-10-30
**It's all in the title**

**Rating:** 4
**Confidence:** 4

**Review:**

The authors propose a clustering methods that builds upon DEC and N2D+tSNE with extra loss terms to preserve geometric structures. They provide an extensive analysis but not always fair. In the end, the benefits of "preserving the geometric structures" are not clear or well defended.

Overall, the paper is well written and easy to follow.
The authors provide arguments to several aspects of their contribution. However it is also scattered with claims.
The maths are clear despite some atypical notations (eg. the scale parameter).

1/The introduction raises well the problem of "learning a representation that favors clustering" (please use \emph instead of bold).
Conversely, the paragraph claiming that we need a clustering that also favors representation learning is not clear. From this point of view, it appears to me to be two completely different things.

2/ The second section reviews the most important and related works in the field of deep clustering.
I would rather describe JULE as a neural extension of hierarchical clustering.
Since I am not an expert in manifold representation, several abbreviation remains unclear to me: LIS, MMD, MLDL, LLE, SSMM.

3/ Since you wish to learn a representation and clustering, you aim to find the optimal \theta AND \mu.
3.1/ The derivations here are the same as DEC (citation), except that you rightfully name the underlying distribution.
"However, we find that the clustering-oriented loss may deteriorate the geometric structure of the latent space"
This claim is not justified.

3.2/ Eq 5. the multiplication dot is unnecessary and disturbing.
Eq 6. The "scale" parameter is very disturbing.
I understand "rank" as the number of clusters. This term ensures that the distance between the manifolds centers in the input space and between the centroids in the features space remain the same/proportional. The separation of the clustering centers depends on their initialization. I wonder what happens when C is larger than the number of true clusters.
"so L rank is easier to optimize, faster to process, and more accurate." Faster to process, yes. However easier and more accurate require more argument. It is not clear relatively to what is it more accurate. Moreover, the conservation of the distances imposed by L_rank  appears to me more difficult to satisfy, especially since the euclidean distance is used in both spaces.

The alignment loss replaces the update of the \mu in DEC.

3.3 It is clear and well explained. Although, Fig 2 and its caption could be more informative and self-contained.

4. The experiments involves several baselines and varied data-sets. However, the experimental setting is not clear and weights a lot in my final opinion.
Do you report best score? Average? What about the baselines? DCLR and N2D reports very similar scores. Are they statistically different? This apply to all the tables.
In the same way that (I)DEC is very dependent on the AE initialization, the results (DCLR vs N2D) suggest the same for the propose method.

4.3.1 Table 3: Most of these measure check if neighbors on the input are also neighbors in the feature space.
Is RMSE relevant since the f is non linear? Note that for LGD, since it stays in the neighborhood, it is not such a big deal.
Why is a high CRA desirable? Besides if DCLR did not report a very high value, that would  suggests that L_rank fails.

4.3.2 "Significantly, the performance of DEC on downstream tasks deteriorates sharply and even shows a large gap with the simplest AEs, which once again shows that the clustering-oriented loss may damage the data geometric structure."
A direct competitor to DCLR is rather IDEC than DEC because of the missing reconstruction loss.
Without statistical test, the differences are not interpretable (SVM).

4.4 Here again, and especially for the last three lines of Table 5, the missing statistical test prevent us from drawing any conclusion.

A.7 It should be say more clearly that the rank of j relatively to i is in terms of distance. It is also related to CRA.
N2D and DCLR report very similar scores on MNIST (Table 1), although the curves on Fig A.3.g are shifted. Hence the argument on the benefit of having a centered curves does not hold.

As a summary, the paper is well written but includes several claims that are not justified. The experimental section is not reproducible (best score? average? same init for all the baselines?) and miss statistical tests.
Indeed some geometric aspects are preserved by the projection, but are relevant for clustering? or classification?
Besides, the benefits over N2D do not appear clear to me. It seams that the method depends on its initialization. I wonder how (I)DEC would perform with the same initialization.

---

> ### Author Response · Authors · 2020-11-24
> **Response to reviewer 1 (#1)**
>
> Thanks for your insightful reviews and we appreciate your valuable suggestions! We address your concerns and questions as follows:
> - Q1: In the end, the benefits of "preserving the geometric structures" are not clear or well defended.
> - Answer1:
> -- **A.** Thanks for your comment. Sorry, we didn't explain it clearly. First, that geometric structure preservation favors clustering was first proposed by IDEC, but they do not provide analysis and evidence to support their claim. Unlike IDEC, which only preserves the local structure based on reconstruction loss, we define two losses for structure preservation - $L_{LIS}$  and $L_{rank}$  - from a multi-manifold learning perspective, achieving more rigorous and formal structure preservation (preserving the local structure while also preserving the global structure) than reconstruction-based method.
> -- **B.** Second, we want to restate the relationship between geometric structure preservation and multi-manifold representation learning to avoid any possible misunderstanding. What we actually addressing here is **unsupervised multi-manifold representation learning**, but due to page limitations and to avoid excessive repetition, we sometimes simplify it as **representation learning**. (In the revised manuscript, all relevant statements have been modified.) Unlike general representation learning, based on the manifold assumption, manifold representations should satisfy the following two main points: **Point (1)** preserving the local geometric structure within each manifold and **Point (2)** ensuring the discriminability between different manifolds. General representation learning concerns only **Point (2)**, which is consistent with the goal of clustering, but **Point (1)** sometimes contradicts with clustering (the contraction has been analyzed in Section 3.3 in the paper). In this paper, we propose an elegant training strategy to mitigate this contradiction so that clustering helps both Point (1) and Point (2).
> -- **C.** In the end, we achieved the following goals related to clustering, geometric structures, and manifold representations: **(1)** **Clustering helps Point (2)** - ensure the discriminability between different manifolds: clear inter-cluster boundaries in Fig. 4, statistical analysis of global structure preservation in Fig. A5(a)-(e); **(2)** **Local structure can be preserved even in the presence of clustering**: better performance for multi-manifold representation learning and downstream tasks (Table A2 and Table A3), the embedding visualizations of the clustering process in Fig. A3, and various cluster shapes (not all deteriorated to spherical) in Fig. 4 ; **(3)** **Geometric structure preservation helps clustering**: better clustering performance in Table 1 and excellent generalizability (Table 2 and Fig. A2). Furthermore, in the ablation experiments in Table A4, we also show the importance of structure-oriented losses to improve clustering and multi-manifold representation learning performance.
>
> ---
> - Q2: The introduction raises well the problem of "learning a representation that favors clustering" (please use \emph instead of bold).
> - Answer2: Thanks for your suggestion, it has been modified in the revised manuscript.
>
> ---
> - Q3: Conversely, the paragraph claiming that we need a clustering that also favors representation learning is not clear. From this point of view, it appears to me to be two completely different things.
> - Answer3: Thanks for your comment. Sorry, this part was not explained very clearly due to page limitations. What we're actually addressing here is the learning of manifold representations, and its two main points are:  **Point (1)** preserving the local geometric structure within each manifold and **Point (2)** ensuring the discriminability between different manifolds. Corresponding to these two points, we designed the $L_{LIS}$ and $L_{rank}$ constraints respectively. For multi-manifold representation learning, most previous work seems to start by assuming that the label of each data point is known, and then designing the algorithm in a supervised way, which greatly simplifies the problem of multi-manifold learning. However, for unsupervised multi-manifold representations learning, it is a challenging problem to decouple the complex cross-over relations and ensure the separability between multiple manifolds.  One natural strategy is to achieve **Point (2)** - ensuring the discriminability through clustering. However, clustering is actually contradictory to **Point (1)** - preserving the local geometric structure (the contraction has been analyzed in Section 3.3 in the paper). Therefore, it is important to alleviate this contradiction so that clustering helps both Point (1) and Point (2) of multi-manifold representational learning. Therefore, our question is **"how to cluster data that favors multi-manifold representation learning?"**, rather than "why clustering is important for representation learning?"

---

> ### Author Response · Authors · 2020-11-24
> **Response to reviewer 1 (#2)**
>
> - Q4:  I would rather describe JULE as a neural extension of hierarchical clustering.
> - Answer4: Thanks for your suggestion. In the paper, we introduced JULE using what had been described in the original paper. Yes, JULE is a neural extension of hierarchical clustering, and it has been added to the revised manuscript.
>
> ---
> - Q5: Since I am not an expert in manifold representation, several abbreviations remain unclear to me: LIS, MMD, MLDL, LLE, SSMM.
> - Answer5: Thanks for your suggestion. All the manifold-related abbreviations you mention have been replaced with their full names or further explanations in the revised manuscript.
>
> ---
> - Q6: Since you wish to learn a representation and clustering, you aim to find the optimal $\theta$ AND $\mu$.
> - Answer6: Yes, in our framework, both $\theta$ and $\mu$ are learnable weights or parameters, and the goal of optimization is to find the optimal $\theta$ and $\mu$. Previously omitted $\mu$, now added.
>
> ---
> - Q7: "However, we find that the clustering-oriented loss may deteriorate the geometric structure of the latent space" This claim is not justified.
> - Answer7:
> -- **A.** Thanks for your comment. First, this claim was first made in the IDEC paper, which says：“the defined clustering loss may corrupt feature space, which leads to non-representative meaningless features and this, in turn, hurts clustering performance.” and “The autoencoder is used to learn representations in an unsupervised manner and the learned features can preserve intrinsic local structure in data.”
> -- **B.** Despite the claim they make, they do not provide any evidence, and their solution is nothing more than preserving local structure through reconstruction loss. In this paper, on the other hand, we show through the visualization of embedding learned by DEC (Fig. 4b), the visualization of the clustering process (Fig. A3), and statistical analysis (Appendix 9) that the geometry of the latent space is indeed disrupted.
> -- **C.** As shown in Fig. A3, in the early stage of training, with clustering-oriented loss $L_{cluster}$, each manifold loses its local structure, and all of them degenerate into spheres (geometric structure of the latent space has **deteriorated**). As the training progresses, with $L_{LIS}$ dominating, the optimization is gradually focused from global to local, with each manifold gradually recovering its original geometric structure from the sphere (geometric structure of the latent space is **recovered**).
>
> ---
> - Q8: Eq 5. the multiplication dot is unnecessary and disturbing.
> - Answer8: Thanks for your suggestion. The multiplication dot has been removed in the revised manuscript.
>
> ---
> - Q9: The "scale" parameter is very disturbing.
> - Answer9: Thanks for the suggestion. The informal parameter $scale$ has been replaced by $\kappa$ in the revised manuscript.
>
> ---
> - Q10:  I understand "rank" as the number of clusters. The separation of the clustering centers depends on their initialization.
> - Answer10:
> -- **A.** Thanks for your comment. The "rank" is not the number of clusters, but the ranking between clusters, measured in terms of distance. In this paper, the separation of cluster centers is dominated by $L_{rank}$, and the parameter $\kappa$ introduced allows us to control the extent of separation between manifolds in a proportional manner.
> -- **B.** The separation of clusters, although related to initialization, does not depend entirely on initialization; At each epoch, we will recalculate the label assignments $s_i$，derive new manifolds $M_{c}$，obtain new manifold centers $v_i^{X}$和 $v_j^{X}$. That is,  we optimize $L_{rank}$ in a dynamic iterative way, rather than by initializing only once and then separating the different clusters based on the initialization results only.
>
> ---
> - Q11:  I wonder what happens when C is larger than the number of true clusters.
> - Answer11: Thanks for your comment. In the original manuscript, the cluster number $C$ is assumed to be a known prior (which is consistent with the assumptions of almost all deep clustering algorithms).  Thus, we provide an additional experiment to explore what happens when $C$ is larger than the number of true clusters. Taking MNIST-test as an example, we present the embedding visualization with an assumed cluster number $C$ being 10, 11, and 12, respectively. We find that when $C$ is larger than the number of true clusters, the data originally belonging to the same cluster will be split, e.g., a cluster is split into two, but different categories of data still maintain clear boundaries and are not mixed together, somewhat similar to hierarchical clustering. The above experiment and analysis have been placed in **Appendix A.7** and Fig. A4.

---

> ### Author Response · Authors · 2020-11-24
> **Response to reviewer 1 (#3)**
>
> - Q12: "so L rank is easier to optimize, faster to process, and more accurate." Faster to process, yes. However easier and more accurate require more argument. It is not clear relatively to what is it more accurate.
> - Answer12: Thanks for your comment. "Easier" does not mean methodologically easier to understand, but rather refers to easier optimization. "More accurate" means that the optimization is less likely to fall into a local optimum solution, and will output better results. Conventional methods for dealing with inter-manifold separation typically impose $L_{sep}$ in Eq. (7) on all data points from different manifolds, which involves $N×N$ point-to-point relationships. This means that each point may be subject to the push-away force from other manifolds, but at the same time, each point has to meet the isometry constraint with its neighboring points. Under these two constraints, optimization is difficult and it is easy to fall into a local optimal solution and output inaccurate results. In contrast, DCRL's $L_{rank}$ constraint does not act directly on the points, but on the clustering centers, which avoids the above problem and makes it easier to optimize. The above explanations have been added to the revised manuscript.
>
> ---
> - Q13: Moreover, the conservation of the distances imposed by L_rank appears to me more difficult to satisfy, especially since the euclidean distance is used in both spaces.
> - Answer13: Thanks for your comment. At each epoch, we will recalculate the label assignments $s_i$，derive new manifolds $M_{c}$，obtain new manifold centers $v_i^{X}$ and $v_j^{X}$. As the training proceeds, the resulting label assignments gradually stabilize and the $d_X\left(v^X_i, v^X_j\right)$ term in Eq. (6) approaches a constant. In this case, Eq. (6) can be satisfied simply by optimizing the learnable clustering centers in the latent space, and the direct use of Euclidean distances in the low-dimensional (10-D) latent space is reasonable and does not suffer from a difficult optimization problem. As shown in Table A2, the CRA metric of DCRL is not only the best on all datasets but also reaches 1.0, which means that the “rank” between different manifolds in the latent space is completely preserved and undamaged, which proves the effectiveness of our global ranking loss $L_{rank}$.
> ---
> - Q14: The alignment loss replaces the update of the $\mu$ in DEC.
> - Answer14: Thanks for your comment. Alignment loss is an auxiliary term to help align learnable cluster centers with their true centers, and we have not used it as a complete replacement for the update of $\mu$ in DEC. In Eq. (4) and Appendix A1, we provide the derivation for the gradient of the three $\mu$-related loss $L_{cluster}$, $L_{rank}$ and $L_{align}$ with respect to $\mu$. This shows that $\mu$ is learnable and can still be updated in our framework by backpropagation.
>
> ---
> - Q15: It is clear and well explained. Although, Fig 2 and its caption could be more informative and self-contained.
> - Answer15: Thanks for your suggestion. In the original manuscript, due to page limitations, we placed Fig. 2 alongside the text. Now we have added more contents to the caption of Fig. 2 and rearranged it in the revised manuscript.
>
> ---
> - Q16: However, the experimental setting is not clear and weights a lot in my final opinion. Do you report best score? Average? What about the baselines? DCLR and N2D reports very similar scores. Are they statistically different? Besides, the benefits over N2D do not appear clear to me.
> - Answer16:
> -- **A.** Thanks for your comments. In this paper, every set of experiments is run 5 times with 5 random seeds, and the results are averaged into the final performance metric (that has been added in the revised manuscript).
> -- **B.** In the original paper of N2D, it is not clear whether the results it provides are the best or the average. In fact, we experimented with different random seeds according to its original experimental setup, and it does not always maintain high accuracy, e.g. on the mnist-full dataset, with some random seeds, the accuracy was only about 0.870 (vs 0.979 it reported). So we guess that N2D has reported the best metrics.
> -- **C.** To be fair, we provide the average performance of N2D under the same random seeds with the same initialization in the paper. It can be seen that the gap between its performance and that of the DCRL is widened.
> -- **D.** Furthermore, we believe that our advantages over N2D are as follows: (1) In a fair comparison, DCRL has a greater advantage over N2D for ACC and NMI, especially on the Fashion-MNIST (0.710 *vs* 0.671), REUTERS-10K (0.836 *vs* 0.784), and HAR (0.845 *vs* 0.797) datasets. (2) we provide better clustering visualization (e.g., Fig. 4); (3) we have better generalizability (e.g., Table 2 and Fig. A2); (4) better performance on manifold representation learning and downstream tasks (e.g., Table A2 and Table A3).

---

> ### Author Response · Authors · 2020-11-24
> **Response to reviewer 1 (#4)**
>
> - Q17: In the same way that (I)DEC is very dependent on the AE initialization, the results (DCLR vs N2D) suggest the same for the propose method.
> - Answer17:
> -- **A.** Thanks for your comments. First, both DEC, IDEC, and N2D rely heavily on their initialization. In the submitted version, we provide the metrics from their original paper or other related work, because there is no detailed description in their original paper as to whether they are using best or average performance.
> -- **B.** For a fairer comparison, we run the released code using the hyperparameters provided in their paper with the same random seeds and initialization, then report their average performance. It can be seen that with the same settings, the N2D performance decreases compared to the original paper, while the DEC and IDEC performance improves considerably, but still falls short of the DCRL.
>
> ---
> - Q18: Is RMSE relevant since the $f$ is non linear?
> - Answer18: Thanks for your comment. RMSE, which—despite its name—，is not related to the reconstruction error of the autoencoder but merely measures to what extent the **distance matrix** of the original space and the latent space coincide. Now, we have changed all "RMSE" to "$d$-RMSE" in the revised manuscript to distinguish them. Besides, the metric $d$-RMSE has been used to evaluate representations or visualizations in papers such as [1,2].
>
> ---
> - Q19: Why is a high CRA desirable? Besides if DCLR did not report a very high value, that would suggests that L_rank fails.
> - Answer19:
> -- **A.** Thanks for your comment. As defined in Appendix A2, CRA (Cluster Rank Accuracy) measures the changes in ranks (in terms of distance) of cluster centers from the input space $X$ to the latent space $Z$. A high CRA metric implies that, in terms of distance, the ranking of closeness between clusters, i.e., the global information in the raw data, is more preserved, which is a fundamental requirement for multi-manifold representation learning and is also beneficial for clustering.
> -- **B.** As shown in Table A2, DCRL outperforms all other methods, especially in the CRA metric, which is not only the best on all datasets but also reaches 1.0. This means that the “rank” between different manifolds in the latent space is completely preserved and undamaged, which proves the effectiveness of our global ranking loss $L_{rank}$.
>
> ---
> - Q20: "Significantly, the performance of DEC on downstream tasks deteriorates sharply and even shows a large gap with the simplest AEs, which once again shows that the clustering-oriented loss may damage the data geometric structure." A direct competitor to DCLR is rather IDEC than DEC because of the missing reconstruction loss.
> - Answer20: Thank you for your comment. We are well aware that DCRL's direct competitor should be IDEC (DEC is just a baseline), but in the sentence you listed, we are comparing AE and DEC (not DCRL and DEC) in order to show that the clustering-oriented loss may damage the data geometric structure. Although this problem was first raised in the paper of IDEC, IDEC just takes it as a truism without providing any analysis or explanation. IDEC claims that the geometry of the data can be preserved through a single reconstruction loss, whereas this paper analyzes the problem of geometry preservation from several perspectives, of which the downstream task (Table A3) is only one, others include representation learning (Table A2), visualization (Fig. 4 and Fig. A3), generalizability (Table 2 and Fig. A2).
>
> ---
> - Q21:  It should be say more clearly that the rank of j relatively to i is in terms of distance. It is also related to CRA.
> - Answer21: Thanks for your suggestion. Both in the definition of CRA and in the description in Appendix A8, we have added that the defined "rank" is measured in terms of distance in the revised manuscript.
>
> ---
> - Q22: Indeed some geometric aspects are preserved by the projection, but are relevant for clustering? or classification?
> - Answer22: Thank you for your comment. Geometric structure preservation is related to the learning task of multi-manifold representation, but we find that geometric structure preservation can be also beneficial for clustering. However, unlike IDEC, which directly uses reconstruction loss to preserve only the local structure, we achieve this from the perspective of geometric structure preservation, preserving both local and global structure, and getting better clustering performance (Table 1), better visualization (Fig. 4) and better generalizability (Table 2 and Fig. A2). See ***Answer 1*** for a more detailed analysis.
>
> ---
> [1] Michael Moor, Max Horn, Bastian Rieck, and Karsten Borgwardt. Topological autoencoders. In Proceedings of the 37th International Conference on Machine Learning (ICML), Proceedings of Machine Learning Research. PMLR, 2020.
>
> [2] Stan Z Li, Zelin Zang, and Lirong Wu. Markov-Lipschitz deep learning. arXiv preprint arXiv:2006.08256, 2020.

---

### Public Comment · ~Wouter_Van_Gansbeke1 · 2020-11-14
**Omission of prior sota in comparisons**

Dear authors and reviewers,

We would like to point out that several relevant papers were omitted from the state-of-the-art comparison in Table 1. In particular we raise the following concerns:

1) Papers [A, B] report higher performance when clustering the MNIST dataset. Unfortunately, these works are not included in the related work or experiments section.

2) Paper [C] is the current state-of-the-art in unsupervised image classification or clustering. In particular, it outperformed the prior state-of-the-art [A] on CIFAR-10, CIFAR-100-20, STL-10 by large margins. Moreover, [C] was applied to the full large-scale ImageNet dataset with 1000 classes. Therefore, we kindly propose the authors to apply their method to the more challenging datasets from [C]. This would provide a more fair, and thorough comparison with recently proposed methods.

We look forward to any future discussions on this.

[A] Xu, Ji, et al. "Invariant information clustering for unsupervised image classification and segmentation."ICCV. 2019.
[B] Jianlong, Chang, et al. “Deep adaptive image clustering“ ICCV. 2017.
[C] Wouter, Van Gansbeke, et al. "Scan: Learning to classify images without labels." ECCV. 2020.

---

### Decision · Program_Chairs · 2021-01-07
**Final Decision**

**Decision:**

Reject

**Comment:**

During discussion, the reviewers acknowledge improvement of the revised version of the paper through author rebuttal and agree with that the paper is overall well written.

However, the novelty of the paper is not strong, and reviewers share the concern that distinction between self supervised learning and deep clustering is not convincing. Also, the concerns (2) raised by the Reviewer #2 is important, which is about the effectiveness of the proposed two-step procedure: first apply t-SNE to transform into two dimensions, followed by running K-means, while the answer is not well  justified. In my opinion, parameter sensitivity should be studied more carefully. The authors mention that $k$ and $\kappa$ have little effect on clustering performance, while this could imply that the associated terms do not have impact on the proposed method. This point should be clarified further.

Overall, the paper is still not ready for publication, I will therefore reject the paper.